# Tools are under-documented: Simple Document Expansion Boosts Tool Retrieval

**Xuan Lu**[1,2,3,*], **Haohang Huang**[3,*], **Rui Meng**[†], **Yaohui Jin**[1], **Wenjun Zeng**[2,3], **Xiaoyu Shen**[2,3,‡]

[1]Shanghai Jiao Tong University
[2]Ningbo Key Laboratory of Spatial Intelligence and Digital Derivative
[3]Institute of Digital Twin, Eastern Institute of Technology, Ningbo
`lux1997@sjtu.edu.cn  xyshen@eitech.edu.cn`

## Abstract

Large Language Models (LLMs) have recently demonstrated strong capabilities in tool use, yet progress in tool retrieval remains hindered by incomplete and heterogeneous tool documentation. To address this challenge, we introduce **Tool-REX**, a new benchmark and framework that systematically enriches tool documentation with structured fields to enable more effective tool retrieval, together with two dedicated models, **Tool-Embed** and **Tool-Rank**. We design a scalable document expansion pipeline that leverages both open- and closed-source LLMs to generate, validate, and refine enriched tool profiles at low cost, producing large-scale corpora with 50k instances for embedding-based retrievers and 200k for rerankers. On top of this data, we develop two models specifically tailored for tool retrieval: Tool-Embed, a dense retriever, and Tool-Rank, an LLM-based reranker. Extensive experiments on ToolRet and Tool-REX demonstrate that document expansion substantially improves retrieval performance, with Tool-Embed and Tool-Rank achieving new state-of-the-art results on both benchmarks. We further analyze the contribution of individual fields to retrieval effectiveness, as well as the broader impact of document expansion on both training and evaluation. Overall, our findings highlight both the promise and limitations of LLM-driven document expansion, positioning Tool-REX, along with the proposed Tool-Embed and Tool-Rank, as a foundation for future research in tool retrieval. [1]

## 1 Introduction

Tool use has emerged as a critical capability of Large Language Models (LLMs), allowing them to interact with external tools and APIs to accomplish complex real-world tasks (Hsieh et al., 2023; Qin et al., 2023; Liu et al., 2024a). This paradigm, often referred to as tool learning (Qin et al., 2024) or tool-augmented agents (Wang et al., 2024), unleashes the power of LLMs to access real-time factual knowledge, perform complex computations, and interact with a vast ecosystem of applications. Within this paradigm, tool selection is particularly crucial. Specifically, tool retrieval—the task of identifying the most relevant tools from large repositories in response to user queries—serves as the gateway to effective tool use. As the number and diversity of available tools grow to thousands of APIs, the challenge of retrieving the right tool at the right time has become a major bottleneck, often hindering reliable utilization.

To assess and improve this capability, several dedicated benchmarks have been developed in recent years, such as ToolBench (Qin et al., 2023), ToolACE (Liu et al., 2024a), MetaTool (Huang et al., 2024c), and ToolRet (Shi et al., 2025b). While these benchmarks have advanced research, they have also highlighted a fundamental challenge: *tool retrieval suffers from a persistent semantic gap between user queries and tool documentation.* Queries are often ambiguous, under-specified, or phrased in ways that do not align with the formal, technical descriptions of tools. This "lower

---

[*]Equal contribution
[†]Working at Google Cloud AI Research. ‡ Corresponding author.

[1]We release our code on `https://github.com/EIT-NLP/Tool-REX`

semantic overlap" places heavy demands on retrieval models, which must bridge the gap between user intent and tool functionality—particularly for out-of-distribution (OOD) queries with diverse or overlapping phrasing. To narrow this gap, the community has explored a range of query/document expansion and augmentation strategies. For example, Re-Invoke (Chen et al., 2024b) augments tool documents with LLM-generated pseudo-queries and extracts latent intents; EASYTOOL (Yuan et al., 2025) restructures verbose tool documentation into concise, standardized forms; MassTool (Lin et al., 2025) formulates retrieval as a multi-task search problem combining tool-use detection and retrieval; and ScaleMCP (Lumer et al., 2025) explores dynamically weighting different document segments to produce more informative embeddings. However, these approaches primarily work around the problem of poor documentation rather than fixing it at the source.

We argue that these efforts, while valuable, overlook the root cause of poor performance: *the tool documentation itself is inherently flawed, suffering from both a lack of standardization and missing critical information*. This lack of standardization is a critical issue; in the ToolRet (Shi et al., 2025b) dataset, for example, we identify at least seven different phrasings for the same type of function (see Appendix A.2), introducing ambiguity that directly complicates retrieval. Furthermore, the documentation is often incomplete, omitting crucial context such as clear "when-to-use" scenarios and operational limitations. This forces LLMs to guess, often leading to the generation of invalid parameters and subsequent tool execution failures (Yuan et al., 2025). This foundational data problem has created a performance ceiling for existing models and, more importantly, has stifled the development of powerful, dedicated tool retrievers and rerankers.

To address this data-centric challenge, we introduce TOOL-REX (**Tool-R**etrieval with **EX**pansion), a new benchmark and framework for systematically enriching tool documentation. We develop a low-cost, LLM-based pipeline that augments raw documentation with structured, high-value fields including *function description*, *when-to-use*, *limitations*, and *tags*. The value of this enriched data is twofold. First, it directly boosts the performance of existing baseline models. Second, it enables the creation of large-scale training corpora, from which we build two specialized models: Tool-Embed, a dense retriever trained on 50k examples, and Tool-Rank, a reranker trained on 200k examples. On TOOL-REX, our models achieve 56.44 $NDCG$@10, 67.81 $Recall$@10, and 56.60 $Completeness$@10, corresponding to improvements of **+10.23**, **+10.29**, and **+9.08** over the MTEB SoTA open-source model.[2]

In summary, our contributions are:

- We design a low-cost tool document expansion pipeline that systematically enriches tool documentation with structured fields (e.g., function description, when-to-use, limitations, and tags), improving completeness, readability, and utility for both retrieval and reranking.

- We introduce **TOOL-REX**, the first benchmark dedicated to document-expanded tool retrieval, built upon 35 well-established tool-use datasets. Alongside the benchmark, we release two large-scale training corpora: *Tool-Embed-Train* (50k examples for retrievers) and *Tool-Rank-Train* (200k examples for rerankers).

- We develop two dedicated models, **Tool-Embed** (retriever) and **Tool-Rank** (reranker), which establish new state-of-the-art performance on TOOL-REX.

- Finally, our work includes a systematic analysis of document expansion's impact on tool retrieval, providing insights for building the next generation of accurate and efficient tool selection systems.

## 2 DOCUMENT EXPANSION FOR TOOL RETRIEVAL

In this section, we introduce TOOL-REX together with two dedicated models. We first discuss the limitations of existing tool documentation in § 2.2, then we provide an overview of its construction pipeline in § 2.3, and analyze the impact of document expansion fields on retrieval performance in § 2.4, finally present two retrieval models specifically developed for this task: Tool-Embed and Tool-Rank in § 2.5.

---

[2]As of September 20, 2025, the state-of-the-art open-source model on the MTEB leaderboard is `Qwen3-Embedding-8B` (see MTEB leaderboard: `https://huggingface.co/spaces/mteb/leaderboard`).

## 2.1 DATASET COLLECTION

TOOLRET(Shi et al., 2025b) serves as the foundation of our dataset construction, as it is currently the most comprehensive benchmark for tool retrieval. It consolidates 35 publicly available tool-use datasets, including ToolBench(Qin et al., 2023), APIBank (Li et al., 2023a), MetaTool (Huang et al., 2024c), and TaskBench, resulting in 7,615 retrieval tasks over 43,215 unique tools. Building upon this unified resource, Tool-REX preserves the original dataset coverage and categorization. Following TOOLRET, all tools are grouped into three categories:

- **Web APIs**: 36,978 tools / 4,916 tasks
- **Code functions**: 3,794 tools / 950 tasks
- **Customized applications**: 2,443 tools / 1,749 tasks

This organization enables controlled evaluation across both in-domain (Web) and out-of-distribution (OOD) settings, as our training data is sourced exclusively from APIGen (Liu et al., 2024b) and ToolBench (Qin et al., 2023) (Web category), ensuring strict separation between training and evaluation without any data leakage, while the test sets additionally include Code and Customized tools. Further details of datasets are provided in Appendix A.1.

## 2.2 LIMITATIONS OF EXISTING TOOL DOCUMENTATION

This breadth of Tool-Ret also introduces substantial heterogeneity across tool documents. As shown in Appendix A.2, the same function can be described in up to seven distinct formulations across sources, yielding heterogeneous—and often inconsistent—representations of semantically equivalent functionality.

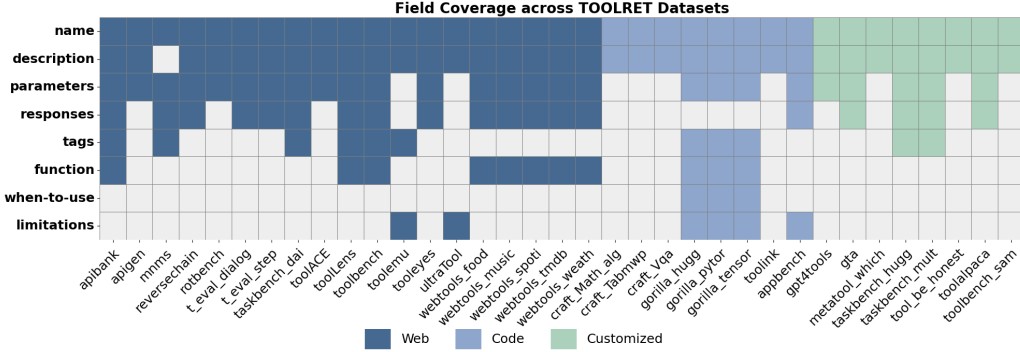

Figure 1: Field coverage across the 35 datasets in TOOLRET.

Beyond inconsistency, many tool documents are incomplete or underspecified. Fig. 1 shows the 35 datasets that constitute the TOOLRET benchmark differ widely in field coverage: some provide relatively rich documentation, while others omit essential components. For example, certain datasets lack critical attributes such as the description field (e.g., MNMS), leaving tools without even a basic textual summary of their functionality. More generally, we observe substantial variation in granularity: some entries include exhaustive parameter lists or implementation notes but omit the tool's operational intent, whereas others present only a broad purpose without clarifying inputs/outputs, preconditions, limitations, or usage contexts.

## 2.3 PROCESS OF TOOL DOCUMENT EXPANSION

TOOL-REX is built upon TOOLRET, which unifies 35 established tool-use datasets categorized into three domains: *Web*, *Code*, and *Customized*. We provide a detailed list of the datasets and their sources in the Appendix A.1.

We construct TOOL-REX through a four-stage pipeline: expansion, judgement, refinement, and human validation, as illustrated in Fig. 2. This design balances cost and quality by combining efficient mid-sized models with stronger models for validation and refinement.

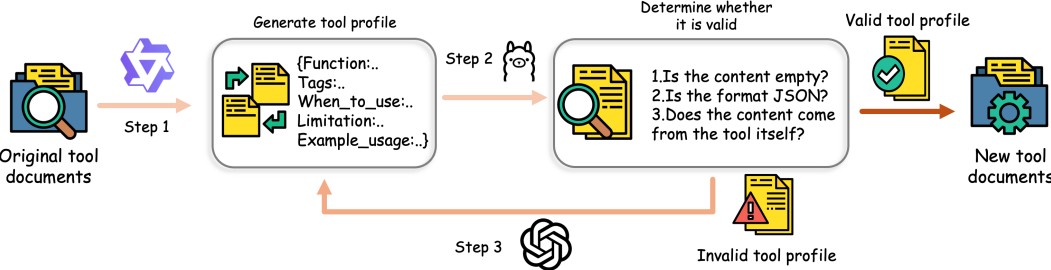

Figure 2: Document expansion pipeline for constructing TOOL-REX.

**Step 1: Expansion.** We use Qwen3-32B, a model with strong instruction-following capability, to expand raw tool documents into structured profiles. During generation, we enable its reasoning mode to improve the quality and consistency of outputs. Given the original documentation $d_{\text{original}}$ as input, the model produces an expanded profile $d_{\text{profile}}$ containing the fields `function_description`, `tags`, `when_to_use`, `limitations`, and `example_usage`. Among them, `function_description` and `tags` are always required, while the other three fields are generated only if explicitly supported by the documentation. The expansion is query-independent and strictly grounded in the tool doc, with unsupported fields omitted. Specifically, `function_description` summarizes the tool's core functionality; `tags` capture key concepts or categories as retrieval keywords; `when_to_use` specifies realistic usage scenarios; `limitations` records constraints such as input/output limits or domain restrictions; and `example_usage` illustrates API calls or queries derivable from the documentation. These structured fields enrich tool docs with semantically precise content, facilitating more effective query–document alignment in retrieval. These structured fields are then encapsulated into a `tool_profile` object $d_{\text{profile}}$, which is merged with the original documentation $d_{\text{original}}$ to form the expanded document:

$$d_{\text{expansion}} = d_{\text{original}} \cup d_{\text{profile}}.$$

The prompt used for generating document expansions is provided in Appendix B.1.

**Step 2: Judgement.** We employ LLaMa-3.1-70B to verify the quality of expanded profiles. Before LLM-based judgement, we first apply rule-based checks to ensure that the generated `tool_profile` is non-empty and conforms to valid JSON format. Subsequently, LLaMa-3.1-70B serves as a semantic judge to assess whether each expanded field is faithful to the original tool documentation, thereby ensuring that no hallucinated or unsupported content is introduced. The prompt used for judgement is provided in Appendix B.2.

**Step 3: Refinement.** For the small fraction of cases that fail Step 2 (about 1.5% of the corpus, roughly 600 examples), we apply a refinement process using GPT-4o, a stronger closed-source model with superior reasoning and consistency capabilities. We reuse the same prompt as in Step 1 to regenerate the expansions, ensuring that the outputs remain strictly grounded in the original tool documentation.

This three-stage process enables efficient and low-cost document expansion, forming the basis of the TOOL-REX benchmark. Examples of generated document expansion are listed in Appendix D.

To verify the quality of the dataset, we conducted sample-based human evaluation to confirm the reliability of the pipeline. We randomly inspected 100 regenerated profiles and checked for faithfulness, absence of hallucinated fields, and semantic consistency. As detailed in Appendix C, all sampled cases were validated by annotators, confirming the robustness of our refinement pipeline.

We further investigate the contribution of each field in Section 2.4. The ablation results show that removing `when_to_use` yields better performance. Based on this finding, we exclude `example_usage` from the expanded profiles used for retrieval, and retain `function_description`, `when_to_use`, `limitations`, and `tags`. Table 1 reports the average token length of tool documents before and after adding the `tool_profile` field.

Table 1: Average token length across domains. We report the original document length, the tool profile length, and the expanded document length.

| Class | Original Doc | ToolProfile | Expanded Doc |
|---|---|---|---|
| code | 167.62 | 51.81 | 219.43 |
| customized | 71.55 | 43.79 | 115.34 |
| web | 156.00 | 42.05 | 198.05 |
| Avg. | 131.72 | 45.89 | 177.61 |

**Training set construction.** Leveraging the outputs from Stage 1 and Stage 2, we construct two large-scale training datasets: approximately 50k instances for training embedding-based retrievers and 200k instances for training rerankers. Importantly, both training sets are derived exclusively from APIGen (Liu et al., 2024b) and ToolBench (Qin et al., 2023), which belong to the **Web** category. Consequently, the models are trained only on Web-based tools, while the evaluation benchmarks additionally include Code and Customized tools that never appear in the training distribution. This setup establishes a genuine out-of-distribution (OOD) evaluation that reflects real-world deployment scenarios, where available tools evolve over time and newly emerging categories must be handled without retraining.

## 2.4 IMPACT OF GENERATED FIELDS ON TOOL RETRIEVAL

Introducing newly generated fields into tool documents is not universally beneficial: longer inputs can dilute salient signals and incur truncation under token budgets retrievers. To quantify these effects, we evaluate two complementary retrievers—BM25 (sparse, keyword overlap) and Qwen3-Embeddings-8B (dense, semantic)—under two protocols using standard IR metrics nDCG@10.

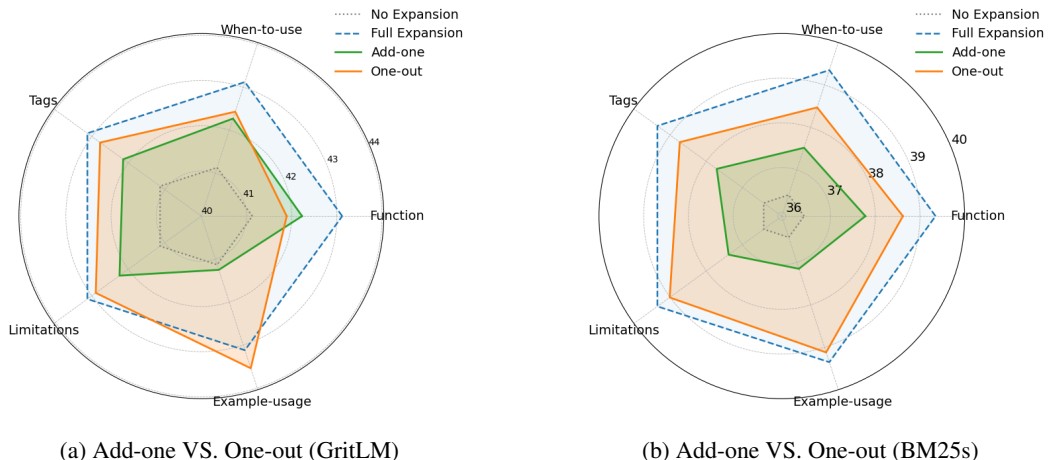

(a) Add-one VS. One-out (GritLM)    (b) Add-one VS. One-out (BM25s)

Figure 3: Impact of individual fields on retrieval: example-usage brings minimal gains in gritlm and bm25s, and even hurts performance in gritlm, while function and when-to-use contribute relatively larger improvements.

**Protocols.** *Add-One:* starting from the original document, we add one generated field at a time from {`function_description`, `when_to_use`, `limitations`, `tags`, `example_usage`} and measure performance deltas. *One-Out:* starting from the fully expanded document, we remove one field at a time and re-evaluate.

We visualize the effect of each field under two protocols for both a dense retriever (GritLM; Fig. 3a) and a sparse retriever (BM25; Fig. 3b). In the plots, the gray dashed ring denotes the *original* baseline, the blue dashed ring denotes *full expansion*, the green polygon traces *Add-one*, and the orange polygon traces *One-out*. Two consistent observations emerge. First, *full expansion* is not uniformly optimal: removing `example_usage` in the One-out protocol yields higher

$N$@10 than keeping it. Second, `example_usage` provides the smallest (often negative) gains in Add-one. In contrast, `function_description` and `tags` are neutral-to-positive under Add-one, and their removal produces noticeable drops in One-out. Guided by these findings, we exclude `example_usage` from the expanded profiles and retain `function_description`, `when_to_use`, `limitations`, and `tags` for retrieval. Appendix D further provides case studies illustrating how expansion improves tool documentation and, in turn, enhances retrieval performance.

## 2.5 ENHANCED MODELS FROM ENRICHED DOCUMENTS

We leverage the synthesized training corpora described in § 2.3 to train our dedicated retriever Tool-Embed and reranker Tool-Rank. All experiments are conducted on two NVIDIA A100 GPUs (80GB each). The training procedures for both components are detailed below.

**Tool-Embed.** For retriever training, we adopt Qwen3-Embedding-0.6B and Qwen3-Embedding-4B as the base model. Training is performed using the `ms-swift`[3] framework with the InfoNCE loss. For each query–tool pair in the 50k training dataset, we randomly sample 5 tools from other queries as negatives. The model is trained for one full epoch with full-parameter tuning using DeepSpeed ZeRO-3.

**Tool-Rank.** For reranking, we build on Qwen3-Reranker-4B using the `LLaMA-Factory`[4] framework, fine-tuned with a cross-entropy objective. We employ LoRA parameter-efficient adaptation, with rank $r = 32$, $\alpha = 64$, and dropout $0.1$. Training is conducted for one epoch on the 200k reranking dataset. At the evaluation phrase, Tool-Rank takes as input a query–tool document pair and is prompted to output whether the document is relevant. A single forward pass yields logits for the tokens `true` and `false`, which are normalized into probabilities:

$$P(\text{relevant} \mid q, d) = \frac{\exp(\ell_{\texttt{true}})}{\exp(\ell_{\texttt{true}}) + \exp(\ell_{\texttt{false}})},$$

where $\ell_{\texttt{true}}$ and $\ell_{\texttt{false}}$ denote the logits of the corresponding tokens. This probability is used as the reranking score. The prompt template used for reranking is provided in Appendix B.3.

To further evaluate the impact of document expansion on tool retrieval, we used the same datasets but removed the expansion field `tool_profile`. With identical training configurations, we trained Tool-Embed_original and Tool-Rank_original as counterparts to Tool-Embed and Tool-Rank.

## 3 EXPERIMENTS

This section first presents the experimental settings in § 3.1, followed by the main results in § 3.2, which include overall comparisons of retrieval models and rerankers.

### 3.1 EXPERIMENTAL SETUP

All experiments are conducted on two NVIDIA A100 GPUs with 80GB memory each. We use the TOOL-REX benchmark described in 2 and ToolRet (Shi et al., 2025b). For reranking experiments, we take the retrieval results from the best-performing retriever in the first stage as the initial set.

**Baselines** We evaluate Tool-Embed-0.6B/4B and Tool-Rank-4B, together with their non-expansion trained counterparts (Tool-Embed_original and Tool-Rank_original). Also, we evaluate 12 representative retrieval and reranking models on Tool-REX:

- Sparse retriever: BM25s (Lù, 2024)
- Dense retriever: GritLM-7B (Muennighoff et al., 2024), NV-Embed-v1 (Lee et al., 2024), gte-Qwen2-1.5B-instruct (Li et al., 2023b), e5-mistral-7b-instruct (Wang et al., 2023a), Qwen3-Embedding series (Zhang et al., 2025): Qwen3-Embedding-0.6B, Qwen3-Embedding-4B, Qwen3-Embedding-8B

---

[3] https://github.com/modelscope/ms-swift
[4] https://github.com/hiyouga/LLaMA-Factory

- LLM-based reranking models: jina-reranker-m0 (Sturua et al., 2024), bge-reranker-v2-gemma (Chen et al., 2024a), Qwen3-Reranker-4B (Zhang et al., 2025)

**Metrics** We adopt three widely used IR metrics to evaluate tool retrieval performance: (i) $NDCG@K$ ($N@K$), which considers both the relevance of retrieved tools and their ranking positions; (ii) $Recall@K$ ($R@K$), which measures the proportion of target tools successfully retrieved within the top-$K$ results; and (iii) $Completeness@K$ ($C@K$) (Qu et al., 2024), which specifically assesses retrieval completeness by assigning $C@K = 1$ if all target tools are included in the top-$K$ results and 0 otherwise.

Table 2: Performance comparison of different retrieval models on ToolRet and TOOL-REX. N@10, R@10, and C@10 denote NDCG@10, Recall@10, and Comprehensiveness@10, respectively.

| Model | Web | | | Code | | | Customized | | | Avg. | | |
|---|---|---|---|---|---|---|---|---|---|---|---|---|
| | N@10 | R@10 | C@10 | N@10 | R@10 | C@10 | N@10 | R@10 | C@10 | N@10 | R@10 | C@10 |
| **ToolRet** | | | | | | | | | | | | |
| BM25s | 26.18 | 34.13 | 22.80 | 41.90 | 56.47 | 55.36 | 41.16 | 48.61 | 38.90 | 36.41 | 46.40 | 39.02 |
| GritLM-7B | 36.58 | 46.01 | 27.65 | 41.26 | 53.81 | 52.07 | 45.55 | 54.01 | 41.40 | 41.13 | 51.28 | 40.37 |
| NV-Embed-v1 | 31.51 | 40.52 | 26.74 | 47.92 | 62.07 | 59.60 | 48.70 | 57.69 | 43.88 | 42.71 | 53.43 | 43.41 |
| gte-Qwen2-1.5B-instruct | 36.73 | 46.39 | 28.24 | 40.56 | 53.08 | 51.37 | 46.52 | 55.38 | 42.10 | 41.27 | 51.62 | 40.57 |
| e5-mistral-7b-instruct | 32.59 | 42.17 | 27.51 | 44.05 | 57.43 | 55.09 | 43.42 | 50.69 | 39.16 | 40.02 | 50.10 | 40.59 |
| Qwen3-Embedding-0.6B | 38.22 | 46.17 | 28.47 | 48.60 | 62.29 | 60.06 | 42.58 | 49.93 | 40.38 | 43.13 | 52.80 | 42.97 |
| Qwen3-Embedding-4B | 40.90 | 50.14 | 31.97 | 53.56 | **71.12** | **69.61** | 42.15 | 50.80 | 40.22 | 45.54 | 57.36 | 47.27 |
| Qwen3-Embedding-8B | 41.61 | 50.23 | 32.56 | 53.43 | 69.21 | 67.29 | 43.59 | 53.13 | 42.73 | 46.21 | 57.52 | 47.52 |
| Tool-Embed$_{original}$-0.6B | 38.57 | 48.05 | 32.39 | 50.11 | 62.22 | 60.52 | 51.77 | 60.17 | 46.28 | 46.82 | 56.81 | 46.40 |
| Tool-Embed$_{original}$-4B | 40.87 | 50.61 | 34.51 | 54.63 | 68.62 | 67.58 | 52.12 | 60.77 | 46.31 | 49.21 | 60.00 | 49.47 |
| **Tool-Embed-0.6B (ours)** | 42.35 | 52.79 | 34.95 | 49.37 | 64.41 | 62.50 | 50.34 | 60.11 | 46.98 | 47.35 | 59.10 | 48.14 |
| **Tool-Embed-4B (ours)** | **45.74** | **55.54** | **37.20** | 54.40 | 69.09 | 67.43 | **54.38** | **63.76** | **48.42** | **51.51** | **62.80** | **51.02** |
| **Tool-REX** | | | | | | | | | | | | |
| BM25s | 28.46 | 35.85 | 24.04 | 45.16 | 58.59 | 57.21 | 44.43 | 50.08 | 38.98 | 39.35↑ | 48.17↑ | 40.08↑ |
| GritLM-7B | 34.46 | 44.01 | 28.96 | 44.39 | 58.08 | 55.94 | 51.79 | 60.17 | 45.82 | 43.54↑ | 54.07↑ | 43.57↑ |
| NV-Embed-v1 | 31.96 | 41.12 | 27.03 | 48.79 | 62.96 | 60.05 | 48.89 | 57.91 | 44.05 | 43.21↑ | 54.00↑ | 43.71↑ |
| gte-Qwen2-1.5B-instruct | 36.09 | 46.60 | 29.21 | 41.39 | 53.10 | 51.09 | 47.87 | 56.12 | 42.81 | 41.78↑ | 51.94↑ | 41.03↑ |
| e5-mistral-7b-instruct | 31.35 | 40.77 | 27.22 | 42.04 | 56.01 | 53.94 | 43.17 | 50.62 | 39.10 | 38.85↓ | 49.13↓ | 40.09↓ |
| Qwen3-Embedding-0.6B | 38.87 | 47.24 | 29.78 | 48.62 | 62.72 | 60.36 | 42.40 | 48.71 | 38.94 | 43.30↑ | 52.89↑ | 43.03↑ |
| Qwen3-Embedding-4B | 40.52 | 48.73 | 30.59 | 53.12 | 68.56 | 67.12 | 43.32 | 51.09 | 40.41 | 45.65↑ | 56.13↓ | 46.04↓ |
| Qwen3-Embedding-8B | 41.94 | 50.61 | 32.78 | 52.38 | 67.40 | 65.46 | 44.29 | 52.47 | 41.87 | 46.23↑ | 56.83↓ | 46.70↓ |
| **Tool-Embed-0.6B (ours)** | 42.41 | 53.07 | 35.06 | 49.89 | 64.11 | 62.26 | 51.99 | 60.29 | 46.12 | 48.10↑ | 59.16↑ | 47.81↑ |
| **Tool-Embed-4B (ours)** | **46.28** | **55.93** | **37.61** | **55.87** | **70.48** | **68.81** | **54.54** | **62.97** | **48.40** | **52.23↑** | **63.13↑** | **51.61↑** |

Table 3: Performance comparison of different reranking models on Tool-REX. All rerankers use the retrieval outputs of Tool-Embed-4B as the initial candidate set, and rerank the top-100 results for each query.

| Model | Web | | | Code | | | Customized | | | Avg. | | |
|---|---|---|---|---|---|---|---|---|---|---|---|---|
| | N@10 | R@10 | C@10 | N@10 | R@10 | C@10 | N@10 | R@10 | C@10 | N@10 | R@10 | C@10 |
| **ToolRet** | | | | | | | | | | | | |
| Tool-Embed$_{original}$-4B | 40.87 | 50.61 | 34.51 | 54.63 | 68.62 | 67.58 | 52.12 | 60.77 | 46.31 | 49.21 | 60.00 | 49.47 |
| + jina-reranker-m0 | 43.82 | 54.10 | 35.10 | 52.13 | 65.88 | 64.89 | 51.44 | 62.49 | 47.27 | 49.13 | 60.82 | 49.09 |
| + bge-reranker-v2-gemma | 43.68 | 53.17 | 34.84 | 52.82 | 67.26 | 65.84 | 51.96 | 62.53 | 47.54 | 49.48 | 60.99 | 49.41 |
| + Qwen3-Reranker-4B | 44.00 | 53.68 | 34.09 | 52.02 | 67.19 | 65.12 | 53.12 | 62.74 | 47.84 | 49.71 | 61.20 | 49.02 |
| + Tool-Rank$_{original}$-4B | 45.46 | 56.02 | 38.66 | 51.75 | 68.21 | 66.35 | 56.42 | 63.19 | 48.91 | 51.21 | 62.47 | 51.31 |
| **+ Tool-Rank-4B (ours)** | **46.25** | **57.21** | **39.96** | **53.01** | **69.19** | **67.57** | **57.85** | **63.38** | **49.67** | **52.37** | **63.26** | **52.40** |
| **Tool-REX** | | | | | | | | | | | | |
| Tool-Embed-4B | 46.28 | 55.93 | 37.61 | 55.87 | 70.48 | 68.81 | 54.54 | 62.97 | 48.40 | 52.23 | 63.13 | 51.61 |
| + jina-reranker-m0 | 46.90 | 56.58 | 38.78 | 54.82 | 70.07 | 69.06 | 56.07 | 65.67 | 50.72 | 52.60 | 64.11 | 52.85 |
| + bge-reranker-v2-gemma | 47.96 | 57.72 | 39.38 | 55.51 | 71.31 | 69.15 | 55.09 | 63.64 | 49.61 | 52.85 | 64.22 | 52.71 |
| + Qwen3-Reranker-4B | 47.48 | 58.12 | 40.84 | 54.16 | 70.33 | 64.48 | 58.66 | 66.45 | 52.34 | 53.43 | 64.97 | 52.55 |
| **+ Tool-Rank-4B (ours)** | **50.66** | **61.36** | **44.45** | **58.69** | **74.00** | **72.05** | **59.97** | **68.05** | **53.29** | **56.44** | **67.81** | **56.60** |

## 3.2 MAIN RESULTS

**Documents Expansion Improve Tool Retrieval**  To assess the contribution of document expansion, we first compare models trained with expansion against non-expansion counterparts under identical settings. As shown in Table 2, relative to the baseline Qwen3-Embedding-0.6B (43.13 $N$@10), Tool-Embed 0.6B/4B yield gains of +4.97/+6.69, whereas the non-expansion-trained Tool-Embed$_{\text{original}}$ 0.6B/4B improve by +3.69/+3.67. Reranking exhibits the same pattern: non-expansion-trained Tool-Rank$_{\text{original}}$ adds +2.00 in $N$@10 after reranking the retrievel result from Tool-Embed$_{\text{original}}$-4B, while the Tool-Rank improves the result of Tool-Embed-4B by +4.21.

We then compare the performance of various retrievers on both ToolRet and TOOL-REX, with results summarized in Table 2. Among the sparse models, BM25s shows consistent improvements on TOOL-REX, with notable gains in $N$@10, $R$@10, and $C$@10, demonstrating that document expansion enhances keyword overlap between queries and tool documents. For dense retrievers, we observe a similar trend: except for E5-Mistral, which slightly decreases in performance, most models achieve higher scores on TOOL-REX. In particular, GritLM achieves the largest gain in $N$@10 (**+2.41**), accompanied by improvements in $R$@10 and $C$@10. For Qwen3-Embedding-4B and Qwen3-Embedding-8B, we note a small decrease in $R$@10 (–1.23 and –0.69, respectively), yet both models still show slight improvements in $N$@10, indicating that expansion helps improve the semantic alignment between queries and tool documents even when recall is marginally reduced. For rerankers, as shown in Fig. 3, when initial results come from Tool-Embed$_{\text{original}}$-4B on ToolRet, gains on the original ToolRet are limited (jina-reranker-m0: –0.08; bge-reranker-v2-gemma: +0.27; Qwen3-Reranker-4B: +0.50), suggesting that under-informative documents constrain reranking; on the document-expanded TOOL-REX, the same rerankers benefit markedly (jina-reranker-m0: +0.37; bge-reranker-v2-gemma: +0.62; Qwen3-Reranker-4B: +1.20), with Tool-Rank-4B providing the largest improvement (+4.21, reaching **56.44**). Collectively, these results show that expansion improves training effectiveness and downstream retrieval quality: it reinforces lexical signals for sparse models and, crucially, supplies additional semantic context that enables dense retrievers and rerankers to capture finer-grained relevance, yielding larger gains than non-expansion training under the same conditions.

**Generalization of Tool-Embed and Tool-Rank**  As shown in Table 2 and Table 3, both Tool-Embed and Tool-Rank demonstrate strong generalization beyond the expanded setting. First, on the non-expanded Tool-Ret benchmark, our models achieve the best performance and even outperform their counterparts trained on the original documentation. This indicates that the enhanced fields in Tool-REX provide effective semantic supervision, enabling models to improve retrieval quality even when evaluated on unmodified documents. Furthermore, the results on the Code and Customized categories confirm OOD generalization. As detailed in Section 2.3, both Tool-Embed-train and Tool-Rank-train are constructed exclusively from Web-category tools, meaning that Code and Customized tools never appear during training. Despite this distribution shift, our models consistently surpass all baselines on these unseen categories, demonstrating that they learn transferable semantic representations rather than overfitting to surface patterns of the training domain.

**SoTA Performance of Tool-Embed and Tool-Rank**  Table 2 shows that our retriever trained on expanded documents, Tool-Embed-4B, achieves state-of-the-art performance on TOOL-REX, reaching $N$@10 = 52.23, $R$@10 = 63.13, and $C$@10 = 51.61. It also surpasses all other retrieval models on the original ToolRet benchmark. Notably, Tool-Embed-0.6B already outperforms baseline models with up to 8B parameters, demonstrating the effectiveness and robustness of the Tool-Embed Table 3 reports results of rerankers on TOOL-REX. All rerankers operate on the initial retrieval results produced by TOOL-EMBED-4B, reordering the top-100 candidates for each query. We observe that rerankers consistently refine retrieval performance, with Tool-Rank-4B achieving state-of-the-art results: $N$@10 improves by **+4.21** (to 56.44), $R$@10 by **+4.68** (to 67.81), and $C$@10 by **+4.99** (to 56.60), compared with the first-stage retriever. These findings confirm the superior performance of Tool-Embed-4B and Tool-Rank-4B on TOOL-REX, establishing strong baselines for future tool retrieval research.

## 4 ANALYSIS

### 4.1 DOCUMENT EXPANSION AS AN EFFECTIVE TRAINING SIGNAL

To isolate the effect of document expansion during training, we construct non-expanded counterparts of our corpora by removing the expansion field `tool_profile` while keeping the number of query–tool pairs and all hyperparameters unchanged. We then train Tool-Embed$_{\text{original}}$ and Tool-Rank$_{\text{original}}$ on these reduced datasets and compare them with the expansion-trained variants. As summarized in Table 2, expansion yields consistently larger gains for retrieval: relative to Qwen3-Embedding-0.6B (43.13 $N@10$), Tool-Embed$_{\text{original}}$-0.6B/4B improve by +3.69/+3.67, whereas expansion-trained Tool-Embed-0.6B/4B achieve +4.97/+6.69. A similar pattern holds for reranking (Table 3): Tool-Rank$_{\text{original}}$ adds +2.00 $N@10$, while expansion-trained Tool-Rank improves by +4.21 to 56.44. Notably, these improvements persist even when evaluated on datasets whose test documents are not expanded (e.g., ToolRet), indicating that exposure to richer, standardized fields during training enhances generalization to heterogeneous documentation. Conceptually, expansion reduces structural heterogeneity (e.g., inconsistent field naming and narrative styles) and supplies explicit functional, situational, and constraint signals (`function`, `tags`, `when_to_use`, `limitation`), which stabilizes optimization and encourages alignment with task-relevant semantics rather than surface cues. Taken together, the evidence supports document expansion as a principled and query-agnostic training signal that improves both embedding and reranking models under fixed training budgets.

### 4.2 DOCUMENT EXPANSION AT EVALUATION TIME: DISCRIMINATIVE GAINS DESPITE SIMILARITY DILUTION

We next examine the role of expansion at evaluation time. To this end, we randomly sampled 100 queries from each of the *Web*, *Code*, and *Customized* tasks (300 in total), and for each query selected one positive and one negative candidate. Using the GritLM model, we computed their similarities with and without document expansion. As shown in Fig. 4, the absolute similarity scores of expanded documents are consistently lower than those of their non-expanded counterparts, plausibly due to semantic dilution caused by increased length. Importantly, the reduction is asymmetric: positives drop only slightly (Web: $-0.0014$, Code: $-0.0022$, Customized: $-0.0027$), whereas negatives show much larger decreases (Web: $-0.0055$, Code: $-0.0038$, Customized: $-0.0152$). This asymmetry yields sharper positive/negative separability and improves relative ordering, which directly benefits retrieval quality.

The effect becomes even clearer at the reranking stage. When Tool-Embed-4B provides the initial pool, rerankers applied on the original, non-expanded ToolRet yield limited or even negative changes in $N@10$ (e.g., jina-reranker-m0: $-0.56$; bge-reranker-v2-gemma: $+0.27$; Qwen3-Reranker-4B: $+0.50$), suggesting that under-informative documents constrain reranker utility. In contrast, rerankers consistently benefit on the expanded Tool-REX: jina-reranker-m0 $(+0.37)$, bge-reranker-v2-gemma $(+0.62)$, Qwen3-Reranker-4B $(+1.20)$, with Tool-Rank-4B achieving the largest gain $(+4.21$, reaching 56.44$)$. A controlled toggle experiment—fixing the retrieval pool while switching between expanded and non-expanded document

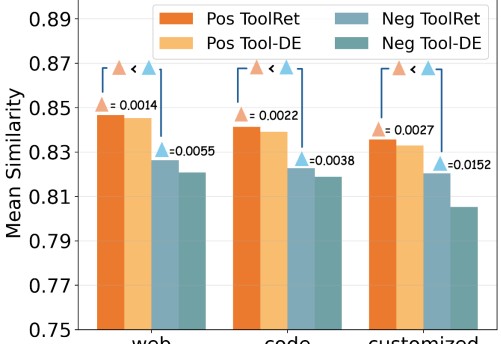

Figure 4: Mean similarity scores of positive and negative in TOOL-REX and TOOLRET.

views—further confirms that removing expansions sharply attenuates reranker improvements. Taken together, these results demonstrate that expansion serves not only as an effective training signal but also as a crucial evaluation-time enabler: it reduces semantic dilution asymmetrically, sharpens the separation between positives and negatives for retrievers, and provides the structured semantic hooks that rerankers require for fine-grained relevance judgments.

## 5 RELATED WORKS

### 5.1 TOOL RETRIEVAL BENCHMARKS

Tool retrieval aims to identify the most relevant tools from large repositories given a user query. To enable systematic evaluation, several tool-use benchmarks have been proposed. ToolBench (Qin et al., 2023) comprises over 16,000 web APIs crawled from RapidAPI with LLM-generated queries and labeled ground truth tools. APIBank (Li et al., 2023a) focuses on web APIs for personalized applications such as alarm booking and database login. ToolACE (Liu et al., 2024a) provides comprehensive evaluation of tool utilization capabilities across diverse scenarios. MetaTool (Huang et al., 2024c) evaluates whether LLMs can correctly decide when and which tools to use. TaskBench (Shen et al., 2024) assesses tool-use capabilities across multimedia, daily life, and deep learning domains. T-Eval (Chen et al., 2023) offers fine-grained evaluation across multiple aspects including instruction following and planning. RestGPT (Song et al., 2023) focuses on connecting LLMs with RESTful APIs for real-world applications. Building upon these resources, ToolRet (Shi et al., 2025b) aggregates over 30 well-established tool-use datasets into a unified format, resulting in 7.6k diverse retrieval tasks and a corpus of 43k tools. Unlike existing benchmarks that manually pre-select small sets of relevant tools, ToolRet evaluates tool retrieval in realistic scenarios with large-scale toolsets, categorizing tasks into three groups: web APIs, code functions, and customized applications.

### 5.2 QUERY AND DOCUMENT EXPANSION

Traditional information retrieval has long explored query and document expansion to reduce noise and enhance semantic matching Li et al. (2025). For instance, Query2doc (Wang et al., 2023b) enriches query representations by generating pseudo-documents, while EnrichIndex (Chen et al., 2025) decomposes documents into multiple semantic views such as summaries, purposes, and QA pairs. Recent advances extend these paradigms to tool retrieval, aiming to bridge the semantic gap between user requests and formal tool documentation. On the query side, MCP-Zero (Fei et al., 2025) converts natural language inputs into structured tool requests for more accurate alignment, and Re-Invoke (Chen et al., 2024b) incorporates an intent extractor to filter tool-relevant intents from verbose queries. For complex or multi-step tasks, TURA (Zhao et al., 2025) performs multi-intent query decomposition, while MassTool (Lin et al., 2025) enriches query representations through search-based user intent modeling with nearest neighbors, improving robustness against out-of-distribution queries. On the document side, EASYTOOL (Yuan et al., 2025) restructures lengthy and inconsistent tool documentation into concise, standardized instructions with usage examples. Re-Invoke also augments tool documents with synthetic queries during offline indexing, and ToolBench (Qin et al., 2023) leverages large-scale LLM prompting to generate diverse instructions for APIs, creating training corpora for retrievers. More advanced strategies enrich documents with topics and intents, as in Advanced RAG-Tool Fusion, or dynamically weight heterogeneous segments (e.g., name, description, synthetic queries) to form discriminative embeddings, as proposed in ScaleMCP (Lumer et al., 2025). PLUTO (Huang et al., 2024b) extends retrieval beyond static expansion by introducing Plan-and-Retrieve and Edit-and-Ground paradigms, combining neural retrieval with LLM-based task decomposition and scenario-driven document edits to enhance tool utilization. Despite these advances, retrieval quality remains a bottleneck in end-to-end tool use. Multiple studies show that retrieval errors propagate and significantly degrade downstream agent performance, underscoring the centrality of retrieval as a foundation for scalable tool-augmented systems.

## 6 CONCLUSION

We present TOOL-REX, the first benchmark and framework dedicated to document-expanded tool retrieval. Through a scalable LLM-based expansion pipeline, we enrich the raw tool documentation with structured fields, producing more complete and consistent resources for training and evaluation. Building on these resources, we introduce Tool-Embed and Tool-Rank which achieve state-of-the-art results on both ToolRet and TOOL-REX. Our analyses show that document expansion not only reduces structural heterogeneity during training but also sharpens positive–negative separability at evaluation, thereby improving reranker effectiveness. We hope TOOL-REX, together with Tool-Embed and Tool-Rank, will serve as a foundation for future research in tool retrieval and tool-augmented intelligence.

## ETHICS STATEMENT

Tool-REX studies LLM-based document expansion for tool retrieval. Our data are collected from publicly available tool-use corpora/APIs, and we do not include sensitive, personal, or proprietary information. Since LLM-generated expansions may contain inaccuracies, we mitigate this risk via automatic validation and spot-checked human inspection. We encourage responsible use and verification of expanded tool descriptions in downstream systems.

## PRODUCIBILITY STATEMENT

We release the Tool-REX benchmark with both original and expanded tool documents, and we will provide code/scripts for preprocessing, training, and evaluation to reproduce all reported results. Implementation details (models, hyperparameters, hardware, and evaluation protocols) are specified in the main paper and supplementary material.

## ACKNOWLEDGEMENTS

We thank EIT and IDT High Performance Computing Center for providing computational resources for this project. This work was supported by the 2035 Key Research and Development Program of Ningbo City under Grant No.2025Z034.

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

## A  DETAILS OF BENCHMARK

### A.1  DATASET CATEGORIZATION

TOOL-REX is built upon ToolRet by applying document expansion. The ToolRet benchmark is one of the most comprehensive benchmarks in the field of tool retrieval, integrating a wide range of well-established tool-use datasets collected from top conferences and large-scale projects. It is divided into three subtasks: (i) **Web APIs**, which are standardized JSON-based APIs (OpenAPI format) that can be invoked via HTTP requests and cover diverse domains such as movies, music, and sports; (ii) **Code Functions**, which consist of source code functions focusing on computational or atomic operations (e.g., tensor calculations, calling Hugging Face models, or using PyTorch libraries); and (iii) **Customized Apps**, which are described in free-form natural language and typically correspond to user-oriented applications such as sending emails or other personalized tasks.

The datasets corresponding to these three subtasks are listed below. For detailed descriptions of each dataset, please refer to their original papers as well as the ToolRet (Shi et al., 2025b).

**Web**  AutoTools-Food (Shi et al., 2025a), RestGPT-TMDB (Song et al., 2023), AutoTools-Movie (Shi et al., 2025a), AutoTools-Weather (Shi et al., 2025a), RestGPT-Spotify (Song et al., 2023), AutoTools-Music (Shi et al., 2025a), ToolBench (Qin et al., 2023), ToolLens (Qu et al., 2024), APIbank (Li et al., 2023a), MetaTool (ToolE) (Huang et al., 2024c), Mnms (Ma et al., 2024), Reverse-Chain (Zhang et al., 2023), ToolEyes (Ye et al., 2024), APIGen (Liu et al., 2024b), UltraTool (Huang et al., 2024a), and T-Eval (Chen et al., 2023);

**Code**  Gorilla-PyTorch (Patil et al., 2024), Gorilla-Tensor (Patil et al., 2024), Gorilla-HuggingFace (Patil et al., 2024), CRAFT-TabMWP (Yuan et al., 2023), CRAFT-VQA (Yuan et al., 2023), and CRAFT-Math-Algebra (Yuan et al., 2023);

**Customized**  ToolACE (Liu et al., 2024a), GPT4Tools (Yang et al., 2023), TaskBench (Shen et al., 2024), ToolAlpaca (Tang et al., 2023), ToolBench-sam (Xu et al., 2023), ToolEmu (Ruan et al., 2023), and TooLink (Qian et al., 2023).

### A.2  CANONICALIZATION OF TOOL-DOCUMENT FIELDS

To enable consistent cross-dataset comparison, we canonicalize tool-document fields by merging different field names with the same semantic meaning into eight unified categories. Table 4 lists the mapping from raw fields observed in the datasets to the canonical categories used in our analysis.

Table 4: Mapping from raw fields (left) to canonical fields (right). Representative raw names are shown; semantically equivalent variants are grouped.

| Raw field names (examples) | Canonical field |
|---|---|
| `name, name_for_human` | name |
| `description, description_for_human, func_description, functionality` | description |
| `category, category_name, domain` | tags |
| `parameters, api_arguments, optional_parameters, required_parameters, inputs, additional_required_arguments, optional_arguments` | parameters |
| `responses, response, return_data, outputs, result_arguments, template_response, output` | responses |
| `method, api_call, url, path` | function |
| `example_usage, example_code` | when-to-use |
| `limitation, is_transactional, performance, python_environment_requirements, doc_arguments` | limitations |

We retain the following eight canonical fields for all visualizations and analyses:

```
You are given a tool document containing fields such as
"name", "description", "parameters", and "responses".

Task: Judge whether the document is sufficiently complete
for retrieval purposes. Specifically, check:
(1) Does it contain a clear functional statement?
(2) Does it provide contextual usage guidance
    (e.g., when to use, typical scenarios, or limitations)?

Answer only with:
true  (if the document is sufficient)
false (if the document is insufficient)
```

Figure 5: Prompt template used for LLM-based completeness auditing of tool documents.

- **name**: The identifier of the tool or function (e.g., API name, function name); the minimal unit for reference and invocation.
- **description**: A natural-language summary of the tool's capability and scope; the primary carrier of semantic intent for retrieval.
- **category**: The topical or functional domain of the tool (e.g., movies, music, vision, finance); used for coarse routing and filtering.
- **parameters**: The input schema required to call the tool (names, roles, or constraints), including variants such as optional/required arguments.
- **responses**: The output or return schema produced by the tool, including response payloads or result fields.
- **method**: The invocation modality or entry point (e.g., HTTP method+URL/path, function signature, or API call string).
- **example_usage**: Illustrative usage examples (queries plus concrete calls/snippets) derived from the documentation.
- **limitations**: Explicit constraints or preconditions noted in the documentation (e.g., transactional restrictions, performance notes, environment requirements).

### A.3 AUDIT OF INCOMPLETE DOCUMENTATION

To systematically assess incompleteness, we performed an LLM-based audit. For each domain (*Web*, *Code*, *Customized*), we randomly sampled 100 tool documents (300 total). The LLM was asked to evaluate whether each document contained (i) a clear functional statement and (ii) contextual usage guidance. A document was flagged as incomplete if either was missing. The audit revealed that **41.6%** of the sampled documents were incomplete, confirming that a substantial portion of the corpus lacks sufficient semantic cues for retrieval.

We repeated the same procedure after applying document expansion (see Sec. 2.3), and found that the incompleteness rate dropped to **23.5%**. This demonstrates that expansion substantially improves coverage of key fields and reduces the proportion of underspecified documents.

## B PROMPTS FOR CONSTRUCTING TOOL-REX

### B.1 DOCUMENTS EXPANSION PROMPT

Fig. 6 shows the prompt used for tool document expansion.

### B.2 JUDGEMENT PROMPT

Fig. 7 shows the prompt used for tool document expansion judgement.

## B.3 Reranking Prompt

Fig. 8 shows the prompt used for tool document reranking, followed by (Lu et al., 2025a;b).

## C Human Annotator and Evaluation Guideline

To ensure the reliability of our dataset, we employed human annotators with relevant technical and linguistic backgrounds. Each annotator was required to have professional experience in software engineering, graduate-level education in computer-related disciplines, and adequate English proficiency for reading and evaluating technical documents. The profile of a representative annotator is shown in Table 5.

Table 5: Background information of a representative human annotator.

| | |
|---|---|
| Name (pseudonymized) | Human Annotator A |
| Educational background | M.Sc. in Computer Technology |
| Professional experience | 3 years in software engineering |
| Language proficiency | Fluent English, IELTS overall band score 6.5 |
| Annotation role | Evaluation of query–document pairs, semantic relevance labeling, and adherence to detailed annotation guidelines |

For the validation step, annotators were provided with detailed evaluation guidelines. Each annotator was asked to assess whether the generated `tool_profile` faithfully and completely reflected the original tool documentation, while avoiding hallucinated or inconsistent content. The full instruction provided to annotators is reproduced below.

---

Human evaluation instruction: You are given a tool's original documentation and a generated `tool_profile`. Please carefully check the following aspects:

- Faithfulness: Do all fields in the `tool_profile` strictly reflect information supported by the original documentation?

- Completeness: Are all important details from the original documentation preserved?

- Hallucination check: Is there any invented functionality, parameter, limitation, or usage example not grounded in the original documentation?

- Consistency: Are the fields logically consistent with each other (e.g., no contradictory descriptions)?

Based on these criteria, decide whether the generated `tool_profile` is acceptable (`Pass`) or not acceptable (`Fail`).

---

## D Examples and Case Study

After adding the `tool profile` field to `tooldoc`, we observed that some tools which were originally outside the top 10 moved into the top 10 with the help of the enhanced `tool profile`. The ranking comparison is based on the `Qwen3-4B-emb` model evaluated on TOOLRET and TOOL-REX.

- **Web Case.** For `t_eval_step_query_21`, the positive tool improved from *outside the top 10* to *rank 4* after adding the `tool_profile` (see Fig. 9).

- **Code Case.** For `craft_Tabmwp_query_55`, the positive tool advanced from *outside the top 10* to *rank 2* after adding the `tool_profile` (see Fig. 10).

- **Customized Case.** For `appbench_query_2`, the positive tool rose from *outside the top 10* to *rank 2* after adding the `tool_profile` (see Fig. 11).

# E  THE USE OF LARGE LANGUAGE MODELS (LLMS)

During the writing process of this manuscript, we utilized `GPT-5` to support language refinement. The model was specifically applied to enhance fluency, readability, and stylistic consistency by correcting grammatical issues and smoothing sentence flow. It is important to note that all conceptual contributions, experimental designs, and results presented in this paper were independently developed and validated by the authors. The LLM's role was strictly limited to linguistic polishing, ensuring that the intellectual content of the work remains fully attributable to the researchers while benefiting from clearer academic presentation.

```
You will receive a copy of the API documentation.
Your tasks are strictly based on this API. A new "tool_profile" field
    has been added to the documentation.
The goal is to make the overall documentation more accurate and
    complete, but not inconsistent.

## Required Fields (Always included in tool_profile)
- "function": less than 20 words;  clearly describe the tool's main
    function using real terms.
- "tags": 3-5 lowercase keywords; no repetition or self-contradictions
    ; phrases covering the main topic, primary operations, and
    synonyms.

## Optional Fields (Included only if explicitly supported by the
    original documentation)
- "when_to_use": less than 20 words; actual usage scenarios.
- "example_usage": Maximum of 2 Include projects only if the actual
    example can be constructed directly from the fields in the
    original document. Each project may contain:
    - "query": A natural or keyword-based question
    - "api_call": An actual API usage call consistent with the tool's
        actual parameters.
- "limitation": Included only if the original document explicitly
    mentions limitations, rate limits, or special conditions.

## Important Rules
- Absolutely no fiction: Do not add functions, parameters, outputs,
    limitations, or fields that do not exist in the original document.
- Consistency: If the original document is about finance, don't
    mention weather. If it's about image processing, don't mention
    stock data. Stick to the relevant fields.
- If optional fields are not explicitly supported, omit them entirely.
- Output only valid JSON objects, formatted as follows:

{
    "tool_profile": {
    "function": "...",
    "tags": ["...", "..."],
    "when_to_use": "...",
    "example_usage": [
    {
        "query": "...",
        "api_call": "..."
    }],
    "limitation": "..."
}}

Important:
- The output must be directly parsable via json.loads() .
- Do not include comments, instructions, or Markdown code fences.
- Do not use the <think> step.
- If the documentation does not explicitly support an optional field,
    omit it rather than adding it yourself.

The "tool_profile" document is now generated based on the following
    API:

{api_document}
```

Figure 6: Prompt template used for tool profile generation.

```
You will receive two inputs:
(1) The original API documentation.
(2) An expanded "tool_profile" derived from the documentation.

Your task is to judge whether the tool_profile
is consistent with the original documentation.

## Field Definitions
- "function": The main purpose of the tool.
- "tags": Keywords describing the domain and usage.
- "when_to_use": Scenarios where the tool should be applied.
- "example_usage": Example queries and corresponding API calls.
- "limitation": Constraints explicitly mentioned in the original
    documentation.

## Important Rule
- Do not invent or assume information.
- Base your judgement only on the given API documentation.

Output only one word: "true" if the tool_profile is consistent,
otherwise "false".
```

Figure 7: Judgement prompt for verifying the consistency of expanded tool_profile with the original API documentation.

```
<|im_start|>system
Judge whether the Tool Document meets the requirements based on the
    Query.
Note that the answer can only be "true" or "false".
<|im_end|>

<|im_start|>user
Query: FILL_QUERY_HERE
Tool Document: FILL_DOCUMENT_HERE
<|im_end|>

<|im_start|>assistant
<think>

</think>
```

Figure 8: The reranking prompt template used in our experiments.

**Web Case**

In the scenario of `t_eval_step_query_21`, the positive tool `t_eval_step_tool_38` rose from **outside the top 10** to rank **4** after the addition of the `tool_profile`.

**Query:** *I am interested in the latest movies in China. Please provide me with the details of the top 3 movies currently playing in China. Additionally, I would like to know the details of the top 5 upcoming movies in China. Finally, I want to get the description of the movie 'The Battle at Lake Changjin'.*

**Query Instruction:** *Given a `movie information retrieval` task, retrieve tools that provide details about currently playing, upcoming, and specific movies by processing parameters like region, number of movies desired, and movie titles to deliver comprehensive movie information aligned with the query's specifications.*

**TOOLRET:**
```
{
  "name": "FilmDouban.coming_out_filter",
  "description": "prints the details of the filtered [outNum] coming
      films in China",
  "required_parameters": [],
  "optional_parameters": [
    {"name": "region", "type": "STRING", "description": "the region of
        search query, must be in Chinese."},
    {"name": "cate", "type": "STRING", "description": "the category of
        search query, must be in Chinese."},
    {"name": "outNum", "type": "NUMBER", "description": "the number of
        search query"}
  ],
  "return_data": [
    {"name": "film", "description": "a list of film information,
        including date, title, cate, region, wantWatchPeopleNum, link
        "}
  ]
}
```

**TOOL-REX** (identical to ToolRet except for the additional `tool_profile` field):
```
{
  ... (same as ToolRet) ...
  "tool_profile": {
    "function": "Filters and retrieves upcoming Chinese films with
        details like title, date, and region",
    "tags": ["film", "upcoming", "china", "filter", "movie details"],
    "when_to_use": "When planning movie outings or researching new
        releases in China",
    "limitation": "Region and category parameters must be provided in
        Chinese"
  }
}
```

Figure 9: Case study: Web query and tool document expansion example.

---

**Code Case**

In the scenario of `craft_Tabmwp_query_55`, the positive tool `craft_Tabmwp_tool_56` rose from **outside the top 10** to rank **2** after the addition of the `tool_profile`.

**Query:** *A business magazine surveyed its readers about their commute times. How many commutes are exactly 43 minutes?*

**Query Instruction:** *Given a `data analysis` task, retrieve tools capable of analyzing commute time data to count instances equal to a specific value, utilizing structured data input according to the query's parameters and criteria.*

**TOOLRET:**
```
{
  "name": "count_instances_with_specific_value_in_stem_leaf(data_frame
      , stem_col, leaf_col, specific_value)",
  "description": "def count_instances_with_specific_value_in_stem_leaf
      (data_frame, stem_col, leaf_col, specific_value):\n \"\"\"\n
      This function takes in a pandas DataFrame representing a stem-
      and-leaf plot of instances and a specific value, and returns the
       number of instances that have values equal to the specific
      value.\n ... (content truncated) ... \n return num_items"
}
```

**TOOL-REX** (identical to ToolRet except for the additional `tool_profile` field):
```
{
  ... (same as ToolRet) ...
  "tool_profile": {
    "function": "Counts occurrences of a specific value in a dataset
        .",
    "tags": ["stem", "leaf", "plot", "count", "frequency"],
    "when_to_use": "Use when you need to count occurrences of a
        specific query in a dataset.",
    "limitation": "Only works with stem-and-leaf formatted data."
  }
}
```

Figure 10: Case study: Code query and tool document expansion example.

---

**Customized Case**

In the scenario of `appbench_query_2`, the positive tool `appbench_tool_21` rose from **outside the top 10** to rank **2** after the addition of the `tool_profile`.

**Query:** *Initiate a private transfer of $93 to Wilson using my credit card, then find an affordable Asian fusion restaurant in San Jose, reserve a table for 4 at the selected restaurant for 11:30 AM on March 3rd, and confirm the reservation.*

**Query Instruction:** *Given a `Transaction and Reservation` task, retrieve tools that facilitate private money transfers and restaurant operations by processing transaction details (such as payment method, amount, and payee) and restaurant-specific information (including location, category, seating, reservation time, and date) to efficiently confirm financial and dining arrangements.*

**TOOLRET:**

```
{
  "is_transactional": true,
  "additional_required_arguments": {
    "payment_method (str)": "The source of funds used for payment.
        Value can only be one of the following: application balance,
        credit or debit card",
    "amount (float)": "The amount to send or request",
    "receiver (str)": "The name of the contact or account making the
        transaction"
  },
  "optional_arguments": {
    "private_visibility (bool)": "Whether the transaction is private"
  },
  "result_arguments": {
    "payment_method (str)": "The source of funds used for payment.
        Value can only be one of the following: application balance,
        credit or debit card",
    "amount (float)": "The amount to send or request",
    "receiver (str)": "The name of the contact or account making the
        transaction",
    "private_visibility (bool)": "Is the transaction private?"
  },
  "name": "makepayment",
  "description": "Send money to a friend"
}
```

**TOOL-REX** (identical to ToolRet except for the additional `tool_profile` field):

```
{
  ... (same as ToolRet) ...
  "tool_profile": {
    "function": "Conveniently send money to a recipient via the API.",
    "tags": ["money transfer", "payment", "api", "transaction", "send
        money"],
    "when_to_use": "When you need to use the API to send money to a
        recipient.",
    "limitation": "Only supports sending money, not receiving money."
  }
}
```

Figure 11: Case study: Customized query and tool document expansion example.

