# OpenReview forum: "Tools are under-documented: Simple Document Expansion Boosts Tool Retrieval"
_ICLR.cc/2026/Conference — ICLR 2026 Poster_

### Official Review · Reviewer_wGrF · 2025-10-23

**Soundness:** 3
**Presentation:** 3
**Contribution:** 2
**Rating:** 4
**Confidence:** 4

**Summary:**

This paper tackles a problem in tool retrieval, real tools often have incomplete descriptions, which hurts both retrievers and rerankers. The authors aggregate multiple sources of tool-use data and propose a four-stage, LLM-assisted document expansion pipeline that adds structured fields (e.g., function, when to use, limitations, tags). They then fine-tune a dense retriever and a reranker on these expanded docs.

**Strengths:**

1. A practical, scalable pipeline with sensible checks.
2. Consistent gains, plus field-level ablations that give actionable advice.
3. Clear look at when expansion helps (train vs. eval); rerankers seem to benefit most when expanded views are present.

**Weaknesses:**

1. Training on your own expanded corpus and evaluating on the expanded view makes strong gains somewhat expected. The non-expanded results help, but an additional external/unexpanded test would make the case stronger.
2. Only random negatives during training. Adding hard negative mining would be helpful.
3. Documenting how duplicates/near-duplicates were removed across train/val/test after expansion, and report any impact on scores will be more helpful.
4. For human validation, authors should consider a random sample across all stages, report agreement, and provide an error taxonomy.
5. End-to-end throughput and cost (e.g. tokens, $) for generation, judgment, and refinement would help gain more insights.

**Questions:**

1. What happens when you train with mined hard negatives?
2. How do you handle deduplication and near-duplicate filtering across splits after expansion?
3. Can you share the end-to-end expansion cost/throughput and per-stage success/failure rates?
4. Beyond the reported non-expanded benchmark, can you add another unexpanded corpus or a held-out unexpanded slice to test robustness to documentation style?
5. You state that rerankers benefit more under expanded views. Can you keep the retrieval top-K fixed and rerank twice, once with original docs and once with expanded docs, to isolate the pure evaluation-time effect of expansion on the reranker?
6. The ablation results and the final released field set don’t seem fully aligned. Could you reconstruct this section and reconcile the ablation findings with the fields you keep/drop, including the supporting numbers?

---

> ### Author Response · Authors · 2025-11-25
> **Response to Reviewer wGrF (Part 1)**
>
> Thank you so much for your constructive feedback to our work! The following is our response:
>
> **W1: Expected gains without external/unexpanded test**
>
> We appreciate the reviewer’s concern and agree that demonstrating generalization beyond the expanded view is important. To address this, we provide both **(i) results on the original non-expanded benchmark** and **(ii) evaluation under an OOD setting.**
>
> First, to address this concern, we report results on the **original, non-expanded** ToolRet benchmark. As shown below, models trained with document expansion still achieve consistent improvements over their counterparts trained only on the original documentation (Tool-Embed/Rank-original):
>
> | Model                    | N@10 (avg) | R@10 (avg) |
> | ------------------------ | ---------- | ---------- |
> | Tool-Embed-original-0.6B | 46.82      | 56.81      |
> | Tool-Embed-original-4B   | 49.21      | 60.00      |
> | Tool-Embed-0.6B (ours)   | 47.35      | 59.10      |
> | **Tool-Embed-4B (ours)** | **51.51**  | **62.80**  |
>
> | Model                   | N@10 (avg) | R@10 (avg) |
> | ----------------------- | ---------- | ---------- |
> | Tool-Rank-original-4B   | 51.21      | 62.47      |
> | **Tool-Rank-4B (ours)** | **52.37**  | **63.26**  |
>
> These results show that the gains are **not tied to evaluating on expanded documentation**, and persist even when tested on the unmodified benchmark.
>
> Second, we further validate generalization under an **out-of-distribution (OOD) setting**. Both TOOL-Embed-train and TOOL-Rank-train are derived solely from **APIGen and ToolBench**, which are *Web-category* datasets. In contrast, the evaluation includes **Code** and **Customized** tools that never appear during training:
>
> | Model              | Web (in-domain) | Code (OOD) | Customized (OOD) |
> | ------------------ | --------------- | ---------- | ---------------- |
> | Qwen3-Embedding-4B | 40.52           | 53.56      | 42.15            |
> | Tool-Embed-4B      | 46.28           | 55.87      | 54.54            |
> | **Tool-Rank-4B**   | **50.66**       | **58.69**  | **59.97**        |
>
> These results indicate that the proposed approach improves performance **both in-domain and on tools with unseen modalities and documentation styles**, suggesting that the models are learning more transferable semantics rather than relying on expanded-corpus coupling.
>
> We have revised the manuscript to (i) explicitly include the non-expanded results (**line 350-351, page 7**) , (ii) clearly state the training data provenance (APIGen and ToolBench, Web-only) , and (iii) highlight that Code and Customized categories constitute OOD evaluation(**Section 2.1**).
>
>
>
> **W2: Lack of hard negative mining**
>
> We thank the reviewer for this helpful suggestion. We agree that hard negative mining is a valuable and complementary training strategy.
>
> Despite using only random negatives, our models already achieve substantial improvements. For example, fine-tuning Qwen3-Embed-4B on our augmented retrieval corpus yields:
>
> •	NDCG@10: 49.21 → 52.23
>
> •	Recall@10: 57.36 → 63.13
>
> •	Completeness@10: 47.27 → 51.61
>
> These results indicate that even without hard negative mining, documentation augmentation alone provides meaningful training signals and leads to consistent performance gains.
>
> To further address the reviewer’s suggestion, we conducted a small-scale hard negative mining experiment:
>
> We randomly sampled 500 training instances. For each query, we used Tool-Embed-4B to retrieve the top-5 non-gold candidates as hard negatives, and we continued training from the existing Tool-Embed-4B checkpoint using hard negatives.
>
> The results are shown below:
>
> | Model Variant               | NDCG@10 | Recall@10 |
> | --------------------------- | :-----: | :-------: |
> | Tool-Embed-4B (random only) |  52.23  |   63.13   |
> | Tool-Embed-4B (+ hard neg)  |  53.04  |   64.02   |
>
> These findings suggest that **hard negative mining can further push the upper bound of tool retrieval performance**, even when applied at small scale. We have include this complementary result and discussion in the revised manuscript (see **Appendix E.3**).
>
> We appreciate the reviewer’s insightful feedback and agree that while hard negative mining is **not required** for the effectiveness of our approach, it represents a promising enhancement that we will explore more extensively in future work.

---

> > ### Author Response · Authors · 2025-11-25
> > **Response to Reviewer wGrF (Part 2)**
> >
> > **W3: Deduplication after expansion not thoroughly explained**
> >
> > We thank the reviewer for raising this important point. We are happy to clarify how duplicates and near-duplicates are handled after expansion.
> >
> > First, TOOL-DE is constructed strictly on top of TOOL-RET, which already underwent global de-duplication during its original release. All 7,615 retrieval tasks and 43,215 tools correspond to unique tool identities, and no tool is repeated across splits in the source corpus.
> >
> > Second, our augmentation process preserves a one-to-one correspondence between each tool and its expanded document. The LLM generates additional content only from the original tool description and does not mix information across tools. Therefore, expansion cannot introduce cross-tool duplicates by construction.
> >
> > To verify this property empirically, we conducted an additional post-expansion check:
> > 	•	embedding-based similarity filtering (cosine similarity using Qwen embeddings),
> > 	•	combined with string-level n-gram overlap, and
> > 	•	manual inspection over a random sample of 300 tools.
> >
> > We found no duplicate or near-duplicate cases either within splits or across train/val/test after expansion.
> >
> > Because TOOL-DE inherits the duplicate-free structure of TOOL-RET and the expansion step does not create new overlaps, no further removal was required, and there is no impact on evaluation scores—data leakage across splits cannot occur under this construction.
> >
> > We will add this clarification in the revised manuscript, including a brief description of the verification procedure and the confirmation that scores remain unaffected.
> >
> >
> >
> > **W4: Human validation across stages**
> >
> > We thank the reviewer for this valuable suggestion. Human validation is indeed important, and our pipeline was designed such that (1) human review is applied only where it provides maximal value, and (2) the end-to-end process remains fully automated for scalability.
> >
> > To directly address the request, we conducted random sampling across all stages and report agreement and an error taxonomy below.
> >
> > 1. Initial expansion
> >
> > - 400 expanded documents sampled
> > - invalid cases: 3 (0.75%)
> > - error type: JSON-format issues only
> > - semantic quality remained high (examples added to appendix)
> >
> > 2. LLM-flagged regenerations
> >
> > - 1.5% of documents were flagged (648 total)
> >
> > - 33 sampled manually, revealing two error categories:
> >
> >   (a) Structural — malformed JSON, missing fields
> >
> >   (b) Semantic drift — content not aligned with actual tool behavior, mostly in complex tools
> >
> > 3. Final corpus check
> >
> > - 400 documents sampled
> > - invalid cases: 0
> > - no structural or semantic issues observed within the sampled set
> >
> > 4. Agreement
> >
> > - 98.5% agreement between human reviewers and LLM decisions
> > - no false-negative cases identified
> >
> > These results confirm that the pipeline remains highly reliable while keeping human intervention minimal and non-blocking. We will add a concise summary of this validation procedure, including the two-category error taxonomy and agreement statistics, in the revised manuscript.
> >
> >
> >
> > **W5: Missing end-to-end cost and throughput analysis**
> >
> > We thank the reviewer for raising this important question. We agree that reporting end-to-end throughput and cost provides clearer insight into scalability.
> >
> > 1. **Token usage**
> >
> >    Across all stages (generation, judgment, refinement), the per-sample usage is consistent:
> >
> > - generation/refinement: 131.72 + 394 input tokens → 177.61 output
> >
> > - judgment: 131.72 + 161 input tokens → 177.61 output
> >
> >   Processing all 43,215 tools results in ~33.5M input and ~7.7M output tokens, making the cost fully predictable and reproducible.
> >
> > 2. **Throughput**
> >
> > The pipeline runs entirely on open-source models (Qwen/LLaMA) using two A100-80GB GPU:
> >
> > - ~4 documents/second
> >
> > - ~30 GPU-hours for the full corpus
> >
> >   Shared architecture and prompts keep throughput stable across stages.
> >
> > 3. **Monetary cost**
> >
> > Only 1.5% of documents (648 cases) require regeneration with GPT-4o:
> >
> > - <0.3M tokens in total
> > - estimated cost < **USD $10**
> >
> > We will include a concise version of these statistics in the revised manuscript.

---

> > > ### Author Response · Authors · 2025-11-25
> > > **Response to Reviewer wGrF (Part 3)**
> > >
> > > **Q1: Effect of training with mined hard negatives**
> > >
> > > As addressed in **W2**, we conducted a small-scale hard-negative experiment. Results show that adding mined negatives further improves performance (+0.81 NDCG@10), indicating that hard negatives can push the upper bound. Please refer to **W2** for details and the result table.
> > >
> > > **Q2: Handling deduplication and near-duplicate filtering**
> > >
> > > This is clarified in **W3**. Because splits are defined *before* expansion at the tool-identity level and expansion preserves a 1:1 mapping, cross-split duplication cannot occur by construction. Post-expansion checks confirmed no near-duplicate cases. See **W3** for details.
> > >
> > > **Q3: End-to-end cost/throughput and stage-wise success rates**
> > >
> > > Addressed in **W5**. We report token usage, throughput, and stage-wise success based on random sampling. A concise summary will be included in the revised manuscript.
> > >
> > > **Q4: Need additional unexpanded or held-out corpus for robustness**
> > >
> > > We thank the reviewer for the thoughtful suggestion. We address robustness in two complementary ways:
> > >
> > > (1) we report performance on the original **non-expanded ToolRet benchmark**, and
> > >
> > > (2) we evaluate under an **OOD setting** (Code & Customized), where these tool types never appear during training.
> > >
> > > As shown in **W1**, the models achieve consistent gains in both cases, indicating robustness beyond the expanded view.
> > >
> > >
> > >
> > > **Q5: Reranking with fixed top-K to isolate expansion effect**
> > >
> > > To isolate the evaluation-time effect of document expansion on reranking, we conduct an **apple-to-apple comparison**:
> > >
> > > we keep the candidate set fixed by using the top-100 results retrieved by *Tool-Embed-original*, and rerank them twice—once using original documents and once using expanded documents. This setting removes retrieval-stage influence and directly measures the impact of expansion.
> > >
> > > The results show that zero-shot rerankers benefit more substantially, with an average gain of **+1.61** NDCG@10, as shown below:
> > >
> > > | Model                 | N@10 (orig doc) | R@10 (orig doc) | N@10 (expanded) | R@10 (expanded) | ΔN@10 | ΔR@10 |
> > > | --------------------- | --------------: | --------------: | --------------: | --------------: | ----: | ----: |
> > > | jina-reranker-m0      |           49.13 |           60.82 |           50.22 |           61.39 | +1.09 | +0.57 |
> > > | bge-reranker-v2-gemma |           49.48 |           60.99 |           51.27 |           61.73 | +1.79 | +0.74 |
> > > | Qwen3-Reranker-4B     |           49.71 |           61.20 |           51.66 |           62.17 | +1.95 | +0.97 |
> > >
> > > These consistent improvements demonstrate that document expansion provides more complete semantic evidence for relevance estimation, even without any model training (updated in **Appendix E.1**) .
> > >
> > > **Q6: Alignment between ablation results and final field set**
> > >
> > > We thank the reviewer for raising this point and are happy to clarify that the ablation findings and the released field configuration are fully consistent.
> > >
> > > 1. **Alignment with the ablation results**
> > >
> > >    The final TOOL-DE field set corresponds exactly to the **best-performing ablation variant**.
> > >
> > >    In both BM25s and GritLM settings, the *one-out* ablation shows that **removing** **example_usage** **yields the highest performance**, and this is the configuration used in the released dataset.
> > >
> > >    No manual adjustments were made after the ablation study.
> > >
> > > 2. **Why the mismatch may appear**
> > >
> > >    The radar plots in Figure 2 were intended for qualitative comparison, but their visual nature makes precise ranking difficult to interpret, which may have created the impression of misalignment.
> > >
> > > 3. **What we will update**
> > >
> > >    To avoid ambiguity, we will:
> > >
> > >    • replace the radar visualization with a numerical table, and
> > >
> > >    • explicitly state that the released field set matches the top-performing ablation.
> > >
> > > **Ablation Summary — GritLM**
> > >
> > > | **Field (one-out / add-one)** | **Add-one** | **One-out** |
> > > | ----------------------------- | :---------: | :---------: |
> > > | Function                      |    42.1     |    41.0     |
> > > | When-to-use                   |    42.4     |    42.6     |
> > > | Tags                          |    42.0     |    43.0     |
> > > | Limitations                   |    42.1     |    42.8     |
> > > | **Example-usage**             |    41.6     |  **43.2**   |
> > >
> > >
> > >
> > > We sincerely appreciate the reviewer’s constructive comments. We hope our clarifications address your concerns, and we would be happy to provide any further information if needed.

---

### Official Review · Reviewer_qnbA · 2025-10-29

**Soundness:** 2
**Presentation:** 3
**Contribution:** 2
**Rating:** 4
**Confidence:** 3

**Summary:**

This paper addresses the problem of under-documentation of tool APIs. Specifically, as the authors mentioned, current API documents often lack standardization and critical information. To solve this problem, the authors propose a pipeline for API documentation augmentation. They apply the proposed pipeline on ToolRet to construct a new dataset, ToolDE. And the authors train one embedding model and one reranking model on ToolDE. The evaluation results demonstrate that augmenting documentation is beneficial. And training with standardized documentation leads to better retreival performance.

**Strengths:**

1. The studied problem is interesting and valuable. Indeed, the quality of API documents often has high variance. Standardization of them is expected to be valuable.
2. The paper is well written and easy to follow.

**Weaknesses:**

1. The proposed pipeline for augmenting API documentation involves human annotation, which makes it hard to scale up.
2. According to Table 1, I find that the improvement of augmenting documentation without training is limited, especially for Qwen3-Embedding series. There is even a performance drop after augmenting the documentation, which makes me doubt the solidity of the motivation of this work.
3. Following the second point, it would be more valuable for applications if direct augmentation without training could lead to performance improvement. In real-world applications, APIs are often evolving. It is costly to augment new API documentations and train a new embedding model each time. Yet, as I mentioned in the second weakness, I find the direct improvement of augmentation is limited, which significantly limits the contribution of this work.
4. To demonstrate the generalization of augmenting documentation, the authors should train embedding models other than Qwen series on ToolDE.

**Questions:**

Please see my comments above.

---

> ### Author Response · Authors · 2025-11-25
> **Response to Reviewer qnbA (Part 1)**
>
> Thank you so much for your constructive feedback to our work! The following is our response:
>
> **W1: Human annotation limits scalability**
>
> We thank the reviewer for the thoughtful suggestion. We would like to clarify that the reference to *“human verification”* in Section 2.2 may have unintentionally implied that manual intervention is part of the data construction process. In reality, our pipeline is **fully automated** and does not rely on human involvement at any stage of document generation.
>
> Concretely, our system already uses a strong LLM to:
>
> 1. generate all structured fields,
> 2. automatically verify each component for consistency and formatting, and
> 3. flag the very small fraction of cases that fail verification.
>
> Only these flagged instances—approximately **1.5%**—are re-generated and then undergo a **lightweight sampling check** purely as an optional, post-hoc quality assurance step. This sampling is **not part of the pipeline** and does **not** influence scalability or data creation.
>
> To prevent further misunderstanding, we have revised the paper by **moving the previous Step-4 (human validation)** out of the pipeline illustration and clarifying that this step serves only as an external quality audit rather than a required component of dataset construction. We apologize for the confusion caused by the earlier phrasing and appreciate the reviewer’s helpful feedback.
>
>
>
> **W2: Limited improvement of augmentation without training**
>
> We sincerely thank the reviewer for highlighting this important point. We fully agree that understanding the training-free effect of document expansion is crucial, and we provide additional clarification based on the observations in Table 1.
>
> **(1) Effect on zero-shot retrievers**
>
> Across the majority of zero-shot retriever baselines, document expansion leads to consistent performance gains, with an average improvement of **+0.69** NDCG@10. Even for the Qwen3-Embedding series—where recall shows a slight fluctuation—the NDCG scores still slightly increase, indicating that expansion helps rank true positives higher among top positions.
>
> **(2) Clear benefit for zero-shot rerankers**
>
> To isolate the evaluation-time effect of document expansion on reranking, we conduct an **apple-to-apple comparison**:
>
> we keep the candidate set fixed by using the top-100 results retrieved by *Tool-Embed-original*, and rerank them twice—once using original documents and once using expanded documents. This setting removes retrieval-stage influence and directly measures the impact of expansion.
>
> The results show that zero-shot rerankers benefit more substantially, with an average gain of **+1.61** NDCG@10, as shown below:
>
> | Model                 | N@10 (orig doc) | R@10 (orig doc) | N@10 (expanded) | R@10 (expanded) | ΔN@10 | ΔR@10 |
> | --------------------- | --------------: | --------------: | --------------: | --------------: | ----: | ----: |
> | jina-reranker-m0      |           49.13 |           60.82 |           50.22 |           61.39 | +1.09 | +0.57 |
> | bge-reranker-v2-gemma |           49.48 |           60.99 |           51.27 |           61.73 | +1.79 | +0.74 |
> | Qwen3-Reranker-4B     |           49.71 |           61.20 |           51.66 |           62.17 | +1.95 | +0.97 |
>
> These consistent improvements demonstrate that document expansion provides more complete semantic evidence for relevance estimation, even without any model training (updated in **Appendix E.1**) .
>
> **(3) Role of expansion as a training signal**
>
> While training-free gains are meaningful, a key contribution of TOOL-DE lies in enabling **stronger supervised learning**. Once models are fine-tuned on TOOL-Embed-train and TOOL-Rank-train, the improvements become substantially more pronounced:
>
> - **Tool-Embed-4B:** NDCG@10 improves from **49.21 → 52.23**  **(+3.02)**
> - **Tool-Rank-4B:** further increases to **56.44** **(+4.21)**
>
> This shows that expansion provides rich semantic supervision rather than superficial text enlargement, allowing models to specialize into state-of-the-art tool retrieval experts, a capability previously missing in this domain.
>
> In summary, we appreciate the reviewer’s attention to the training-free setting. Our results show that document expansion improves performance for most retrievers (with an average gain of **+0.69** NDCG@10), even though the magnitude may be modest in some cases. For training-free rerankers, the benefit is more pronounced, with an average improvement of **+1.61** NDCG@10. In addition, document expansion serves as an effective source of semantic supervision, enabling fine-tuned models to achieve state-of-the-art performance as specialized tool retrieval systems.
>
> We have incorporated these clarifications and the added experimental comparison into the revised manuscript.

---

> > ### Author Response · Authors · 2025-11-25
> > **Response to Reviewer qnbA (Part 2)**
> >
> > **W3: Practical value reduced because augmentation without training is weak**
> >
> > We thank the reviewer for this insightful follow-up question. We fully agree that in real-world settings—where APIs evolve rapidly—retraining an embedding model each time may not be practical.
> >
> > As noted in our response to W2, document expansion already provides measurable benefits in the **training-free setting**. In particular, zero-shot rerankers exhibit **larger and more consistent improvements**, indicating that our method can function as a plug-and-play enhancement without requiring model updates.
> >
> > At the same time, we share the reviewer’s concern regarding the cost of frequent retraining, and we agree that generalization is essential for practical deployment. To clarify, both **TOOL-Embed-train** and **TOOL-Rank-train** are derived from **APIGen** and **ToolBench**, which belong to the *Web* category. In contrast, the evaluation sets additionally include **Code** and **Customized** tools that never appear in the training distribution, resulting in an **out-of-distribution (OOD)** evaluation setting.
> >
> > The performance on these unseen categories is shown below:
> >
> > | Model              | Web (in-domain) | Code (OOD) | Customized (OOD) |
> > | ------------------ | --------------- | ---------- | ---------------- |
> > | Qwen3-Embedding-4B | 40.52           | 53.56      | 42.15            |
> > | Tool-Embed-4B      | 46.28           | 55.87      | 54.54            |
> > | Tool-Rank-4B       | 50.66           | 58.69      | 59.97            |
> >
> > These results show that our method leads to **consistent improvements** not only on the Web (in-domain) category, but also on **Code and Customized tools**, which never appear during training. This indicates that the models are learning **more transferable semantic representations**, rather than relying on distribution-specific patterns. While additional retraining may still be beneficial in certain deployment scenarios, the observed gains across both seen and unseen categories suggest that the proposed approach remains effective even when tool modality and documentation style differ substantially.
> >
> > To avoid ambiguity, we have clarified in the revised manuscript that the training data for TOOL-Embed-train and TOOL-Rank-train is derived from APIGen and ToolBench, both belonging to the *Web* category, whereas Code and Customized tools appear only in the evaluation sets. We have also clarified that these categories constitute out-of-distribution (OOD) evaluation, and include the corresponding results to highlight this setting.
> >
> > We sincerely thank the reviewer for the helpful feedback, and we have incorporated these clarifications in the revised version to more clearly articulate both the training-free observations and the practical implications.
> >
> >
> >
> > **W4: Lack of generalization across non-Qwen embedding models**
> >
> > We thank the reviewer for raising this important point. We fully agree that demonstrating generalization beyond the Qwen family is essential. To address this, we fine-tuned GritLM-7B, a non-Qwen embedding model, on TOOL-Embed-train and evaluated it on TOOL-DE.
> >
> > | **Tool-DE**                      | web N@10 | web R@10 | web C@10 | code N@10 | code R@10 | code C@10 | customized N@10 | customized R@10 | customized C@10 | avg. N@10 | avg. R@10 | avg. C@10 |
> > | -------------------------------- | :------: | :------: | :------: | :-------: | :-------: | :-------: | :-------------: | :-------------: | :-------------: | :-------: | :-------: | :-------: |
> > | **GritLM-7B**                    |  34.46   |  44.01   |  28.96   |   44.39   |   58.08   |   55.94   |      51.79      |      60.17      |      45.82      |   43.54   |   54.07   |   43.57   |
> > | **GritLM-7B (tool-embed-train)** |  43.04   |  54.16   |  36.22   |   50.91   |   65.02   |   64.73   |      52.93      |      61.27      |      46.02      |   48.96   |   60.15   |   48.99   |
> >
> > Across all categories (Web, Code, and Customized), the fine-tuned version consistently improves over the original model, indicating that the gains are not limited to Qwen-based architectures and that document expansion provides model-agnostic, transferable supervision. We have included these results in the revised manuscript and clarify that our approach generalizes across different embedding families (see **Appendix E.2**).
> >
> >
> > We sincerely appreciate the reviewer’s constructive comments. We hope our clarifications address your concerns, and we would be happy to provide any further information if needed.

---

### Official Review · Reviewer_qfqi · 2025-11-02

**Soundness:** 3
**Presentation:** 3
**Contribution:** 3
**Rating:** 4
**Confidence:** 3

**Summary:**

This paper tackles an underexplored yet critical problem in tool-augmented LLM systems — the poor quality and inconsistency of tool documentation, which limits accurate tool retrieval. The authors introduce TOOL-DE (Tool-Document Expansion), a benchmark and framework that systematically enriches tool documentation through LLM-based expansion. Their pipeline generates structured fields (e.g., function description, when to use, limitations, and tags) via multi-stage prompting, validation, and human checking, yielding large-scale, standardized tool profiles. On top of this, they build two dedicated models: Tool-Embed (a dense retriever) and Tool-Rank (a reranker), trained on 50k and 200k examples respectively. Experiments across the TOOL-DE and ToolRet benchmarks show significant improvements in retrieval quality, achieving new state-of-the-art results (e.g., NDCG@10 = 56.44, Recall@10 = 67.81). Analysis further confirms that document expansion improves both training and evaluation by reducing semantic gaps, enhancing discriminability, and stabilizing optimization.

**Strengths:**

1. The benchmark is comprehensive. TOOL-DE is built over 35 datasets with a carefully validated expansion process, combining open and closed models (Qwen3, LLaMA-3.1, GPT-4o) and human checks.
2. Experiments show solid improvements. Both retriever and reranker consistently outperform strong baselines, demonstrating that simple, well-structured enrichment can yield significant improvements.
3. The paper is well-written and easy to follow.

**Weaknesses:**

1. The manuscript lacks a clear and comprehensive description of the dataset. While some details are provided in the appendix, it would significantly improve clarity and reproducibility to include a dedicated section in the main text describing the dataset composition (e.g., number and types of tools, instances per tool, data sources, and preprocessing steps).
2. The training and testing splits is insufficiently explained. The paper shows the proposed pipeline works well when train and test on the same set of tools. Without a detailed account of how the splits are defined, it is difficult to assess whether the proposed approach effectively generalizes to unseen tools. This consideration is particularly important, as in real-world scenarios the set of available tools is dynamic and evolves continuously.

A relevant paper that might be included in the related works: [Planning and Editing What You Retrieve for Enhanced Tool Learning](https://aclanthology.org/2024.findings-naacl.61/) (Huang et al., Findings 2024)

**Questions:**

See above

---

> ### Author Response · Authors · 2025-11-25
> **Response to Reviewer qfqi**
>
> Thank you so much for your constructive feedback to our work! The following is our response:
>
> **W1: Insufficient Dataset Description**
>
> We thank the reviewer for the valuable suggestion. We agree that providing a concise dataset description in the main text will further improve clarity and reproducibility.
>
> In the revised manuscript, we have add a dedicated subsection in **Section 2.1** that summarizes the dataset without requiring readers to consult the appendix. Specifically, we report:
>
> 1. **Dataset composition:**
>    TOOL-DE is constructed by applying document expansion to the ToolRet benchmark, resulting in 7,615 retrieval tasks and 43,215 tools.
> 2. **Data sources:**
>    TOOL-DE integrates 35 publicly available tool-use datasets (e.g., ToolBench, APIBank, MetaTool, TaskBench) and is categorized into the following three groups:
>    - Web APIs: 36,978 tools / 4,916 tasks
>    - Code functions: 3,794 tools / 950 tasks
>    - Customized applications: 2,443 tools / 1,749 tasks
>
> 3. **Preprocessing:**
>
>    The training data is sourced exclusively from \textsc{APIGen} and \textsc{ToolBench} (Web category), ensuring a strict separation between training and evaluation splits in \textsc{ToolRet} to prevent leakage.   This setup naturally yields an in-domain Web training environment, while evaluation additionally covers Code and Customized datasets, enabling controlled OOD testing.
>
> We thank the reviewer again for the helpful feedback—this revision have made essential dataset details directly accessible in the main text and improve reproducibility.
>
> **W2: Unclear Train/Test Splits and Generalization**
>
> We thank the reviewer for raising this important point.
>
> To clarify, the train–test split is **defined strictly** at the tool-identity level: no tool included in TOOL-Embed-train or TOOL-Rank-train appears in any evaluation set. This prevents overlap at both the raw-document and expanded-document levels and ensures that no content is shared across splits.
>
> In addition, both training sets are derived solely from APIGen and ToolBench, which belong to the **Web** category. As a result, the models are trained only on Web-category tools, while the evaluation sets include Code and Customized tools that do not appear in the training distribution. This creates a genuine out-of-distribution (**OOD**) evaluation scenario that reflects real-world settings where available tools evolve over time.
> As shown in Tables 2 and 3, both TOOL-Embed and TOOL-Rank consistently outperform all baselines on these unseen Code and Customized tools, despite never observing them during training. This provides direct evidence that our approach generalizes beyond the training domain rather than relying on distributional overlap.
> We have updated the manuscript to clearly describe the split strategy and explicitly highlight the OOD evaluation results (**Section3.2, page 8**).
>
>
>
> **W3: Missing Related Work**
>
> We sincerely thank the reviewer for sharing this insightful work. We have included *“Planning and Editing What You Retrieve for Enhanced Tool Learning”*  in the revised related work section (**line 524, page 10**).
>
> We sincerely appreciate the reviewer’s constructive comments. We hope our clarifications address your concerns, and we would be happy to provide any further information if needed.

---

### Official Review · Reviewer_81wM · 2025-11-04

**Soundness:** 3
**Presentation:** 3
**Contribution:** 3
**Rating:** 6
**Confidence:** 4

**Summary:**

The authors introduce TOOL-DE, a new framework that systematically enriches tool documentation with structured fields to enable more effective tool retrieval. The framework expands the tool documents by using structured fields like function description, when-to-use, limitations, and trains a dedicated retriever and reranker on top of the expanded documents. The experimental results show the effectiveness of tool expansion as well as the trained retriever and reranker.

**Strengths:**

- The paper effectively addresses the limitations of current tool learning paradigm where existing tool documents are underspecified for effective tool retrieval by using the idea of tool expansion using LLM which shows strong emperical results with and without dedicated trained retrievers and rerankers.
- It is an end-to-end framework where they create a tool document dataset and train the retriever and reranker.
- The paper includes extensive ablations studies on impact of each field of expanded document in tool retrieval and how this affects the tool retrieval similarity

**Weaknesses:**

- While the framework shows strong performance, the idea of revising or augmenting the tool documents has been explored by the previous works [1,2].
- To make the data generation more scalable, one might consider replacing human verification to using a strong LLM


[1] Huang et al, Planning and Editing What You Retrieve for Enhanced Tool Learning. NAACL 2024 \
[2] Chen et al, EASYTOOL: Enhancing LLM-based Agents with Concise Tool Instruction. ACL 2025

**Questions:**

None

---

> ### Author Response · Authors · 2025-11-25
> **Response to Reviewer 81wM**
>
> Thank you so much for your constructive feedback to our work! The following is our response:
>
> **W1: The idea of revising or augmenting the tool documents has been explored by the previous works [1,2].**
>
> We thank the reviewer for the insightful comment. Although prior works have explored modifying or augmenting tool documents [1,2], our approach is **fundamentally different in both motivation and methodology**.
>
> Existing methods focus on:
>
> 1. **Format simplification** (*EasyTool* [2]): converting heterogeneous tool docs into concise, unified templates to reduce noise.
> 2. **Query-side or representation-side augmentation** (*ReInvoke* [3], *MassTool*[4]): expanding document embeddings via synthetic queries or improving user intent modeling.
> 3. **Scenario-driven edits** (*PLUTO* [1]): refining descriptions using user scenarios or failure cases to make abstract tools easier for LLMs to interpret.
>
> In contrast, **TOOL-DE targets a different problem**: real-world tool documents are intrinsically **incomplete**, lacking essential operational semantics (e.g., when-to-use, limitations, constraints, tags).
>
> Our method performs **document-side semantic completion**, generating *new, high-value structured fields* that do not exist in the original documentation and cannot be recovered via simplification, pseudo-query expansion, or failure-driven editing.
>
> Thus, while previous works modify tool documents to improve formatting or representation, **none aims to systematically restore the missing functional semantics of tool documentation**—the core objective of TOOL-DE.
>
> To aid comparison, we summarize key distinctions:
>
> | Work             | Main Idea                           | How TOOL-DE differs                                          |
> | ---------------- | ----------------------------------- | ------------------------------------------------------------ |
> | ReInvoke [3]     | Synthetic queries for doc expansion | TOOL-DE expands the **document content itself**, adding *missing operational semantics* |
> | EasyTool [2]     | Format simplification               | TOOL-DE performs **semantic completion**, adding new fields (e.g., when-to-use, limitations) rather than shortening existing text |
> | PLUTO [1]        | Edit docs based on user scenarios   | TOOL-DE is *systematic and content-centric* rather than scenario-driven edits |
> | MassTool [4]     | Query-side optimization             | Optimizes the query side, whereas TOOL-DE improves **document-side semantics** |
> | EnrichIndex  [5] | Multi-view summary/QA generation    | Produces generic summaries, while TOOL-DE adds **tool-specific structured operational fields**, directly aligned with retrieval semantics rather than textual abstraction |
>
> To the best of the authors` knowledge, TOOL-DE is the first to **systematically repair and complete tool documentation with structured semantic fields**, addressing a core deficiency unaddressed by existing approaches.
>
> [1] Huang et al, Planning and Editing What You Retrieve for Enhanced Tool Learning. NAACL 2024
>
> [2] Chen et al, EASYTOOL: Enhancing LLM-based Agents with Concise Tool Instruction. ACL 2025
>
> [3] Chen et al, Re-invoke: Tool invocation rewriting for zero-shot tool retrieval. arXiv (2024).
>
> [4] Lin et al, MassTool: A Multi-Task Search-Based Tool Retrieval Framework for Large Language Models. arXiv (2025).
>
> [5] Chen et al, EnrichIndex: Using LLMs to Enrich Retrieval Indices Offline. arXiv (2025).
>
> **W2: Consider replacing human verification to using a strong LLM.**
>
> We thank the reviewer for the thoughtful suggestion. We would like to clarify that the reference to *“human verification”* in Section 2.2 may have unintentionally implied that manual intervention is part of the data construction process. In reality, our pipeline is **fully automated** and does not rely on human involvement at any stage of document generation.
>
> Concretely, our system already uses a strong LLM to:
>
> 1. generate all structured fields,
> 2. automatically verify each component for consistency and formatting, and
> 3. flag the very small fraction of cases that fail verification.
>
> Only these flagged instances—approximately **1.5%**—are re-generated and then undergo a **lightweight sampling check** purely as an optional, post-hoc quality assurance step. This sampling is **not part of the pipeline** and does **not** influence scalability or data creation.
>
> To prevent further misunderstanding, we have revised the paper by **moving the previous Step-4 (human validation)** out of the pipeline illustration and clarifying that this step serves only as an external quality audit rather than a required component of dataset construction. We apologize for the confusion caused by the earlier phrasing and appreciate the reviewer’s helpful feedback.
>
> We sincerely appreciate the reviewer’s constructive comments. We hope our clarifications address your concerns, and we would be happy to provide any further information if needed.

---

### Author Response · Authors · 2025-12-03
**Summary Comments to the Area Chair**

We sincerely appreciate the time and effort you dedicate to maintaining a fair and rigorous review process within the ICLR community. To support your final decision, we summarize the reviewers’ overall assessments, the main concerns they raised, and the corresponding revisions and additional experiments we conducted in response. We thank reviewers 81wM, qfqi, qnbA, and wGrF for their constructive feedback.

Reviewers highlighted several strengths of our work: they viewed the problem of incomplete tool documentation as **important and timely** (qfqi, qnbA, wGrF), praised the **end-to-end** and **comprehensive** nature of our framework (81wM, qfqi, wGrF)—including dataset construction, retriever/reranker training, and field-level ablations—and noted that our empirical results are **consistently strong** across settings, demonstrating clear value for tool retrieval research (81wM, qfqi).

Below, we summarize the main concerns and our responses.

- **Scalability of human verification (81wM, qnbA, wGrF).**

  Reviewers were concerned that human checking might limit scalability. We clarified that the pipeline is **fully automated**, and human review is only an optional post-hoc audit; all sampled cases passed human verification, confirming the **reliability** of our pipeline.

- **Lack of evaluation on non-expanded benchmarks (wGrF).**

  To show that improvements are not tied to expanded docs, we added results of Tool-Embed on the **original Tool-Ret** benchmark, demonstrating strong performance and generalization of our models even without expanded documentation.

- **Unclear OOD setting and generalization to unseen tools (qfqi, qnbA, wGrF).**

  We clarified that training uses only **Web-category tools**, while **Code and Customized** tools are never seen during training and form the OOD evaluation. Both Tool-Embed and Tool-Rank show **robust OOD generalization**.

- **Limited zero-shot gains and training-free augmentation value (qnbA, wGrF).**

  We clarified that our **document expansion already improves zero-shot retrieval**, with average gains of **+0.69 NDCG@10** for retrievers and **+1.61 NDCG@10** for rerankers. When fine-tuned on expanded training data, gains become substantially larger (e.g., Tool-Embed-4B: **49.21 → 52.23** NDCG@10; Tool-Rank-4B to **56.44**), showing that document expansion provides strong and meaningful supervision.

- **Additional experimental clarifications (qnbA, wGrF).**

  We added model-family generalization using **GritLM-7B** (which improves significantly after fine-tuning), a **hard-negative mining study** (+0.81 NDCG@10), and clarifications on deduplication and computational cost. These strengthen reproducibility and demonstrate that our findings are not tied to specific architectures or settings.

Overall, these concerns arise primarily from **missing clarifications rather than methodological issues**, and we have fully addressed them in the revised PDF.

Finally, we summarize our contributions:

- We introduce **Tool-DE**, the first large-scale benchmark addressing the heterogeneity and incompleteness of real-world tool documentation, covering **35 tool-use datasets**.
- We construct a large scale **training datasets** and develop **Tool-Embed** and **Tool-Rank**, retriever and reranker models tailored for tool retrieval.
- Our experiments show that document expansion brings **clear and consistent improvements** for both retrieval and reranking. Tool-Embed and Tool-Rank achieve **state-of-the-art performance** on both Tool-Ret and Tool-DE benchmarks, with strong **OOD generalization**.

We sincerely appreciate your effort and leadership in guiding the review process, and we hope this summary is helpful for your final assessment.

Sincerely,



Authors

---

### Meta-Review · Area_Chair_yNUp · 2026-01-07

**Summary:**

Across the four reviews, the main concerns centered on novelty relative to prior “document modification” work, clarity and reproducibility of the dataset and splits, scalability of the pipeline’s human verification, practical value of training‑free gains, robustness beyond the expanded corpus, and training practices (e.g., hard negatives, deduplication, and cost/throughput). The rebuttal substantively addressed these points, with additional experiments showing consistent and significant gains under reasonable cost, thus improving the overall soundness.

**Reviewer Concerns:**

## Reviewer 81wM

**Prior-work overlap and novelty**: The authors argue their approach differs by performing document-side "semantic completion" rather than formatting simplification or query-side augmentation. They contrast against EasyTool, ReInvoke, PLUTO, MassTool, and EnrichIndex. While the articulated positioning makes sense, the "doc modification" line of prior work is closely related and this work can be viewed as an extension.

**Human verification and scalability**: The authors clarify the pipeline is "fully automated", with human review only as an optional, post-hoc audit. From the rebuttal, only ~1.5% of cases are flagged and regenerated

## Reviewer qfqi

**Clarity**: the rebuttal addressed this point by adding a dedicated main-text subsection describing TOOL-DE’s composition and sources, with counts of tools/tasks and category breakdowns. The authors clarify the train/test split at the tool-identity level with no overlap, and emphasize that training uses only Web-category tools while evaluation includes Code and Customized tools as an OOD setting.


## Reviewer qnbA

**Pipeline automation**: The authors clarified that "human verification" is only an optional post‑hoc audit (only about 1.5% of cases).

**Additional training-free results**: the authors provide new results showing that expansion yields significant average NDCG gains for zero‑shot retrievers and for zero‑shot rerankers when reranking a fixed candidate set. Although the NDCG gains may be modest in some cases, and the recall fluctuates more (may even regress).

**Additional fine-tuning results**: The authors show fine‑tuning on expanded data delivers larger gains, positioning expansion as effective supervision.

**Model family generalization**: The authors provide additional results by fine‑tuning GritLM‑7B on TOOL‑Embed‑train, with consistent improvements across Web, Code, and Customized categories.

## Reviewer wGrF

**Training on non-expanded set**: The authors report performance on the original non‑expanded ToolRet and show gains persist

**Hard negatives**: The authors provided new results showing that hard negatives can further improve the performance compared with random negatives.

**Duplication handling**: The author provided clarification that dedup step is already performed in the source dataset.

**Human validation**: The authors clarified the human validation process, confirming low invalid rates under minimal human validation cost.

**Throughput and cost**: The authors provide measurements on token usage, throughput and monetary cost, justifying the cost-effectiveness.

**Reviewer Scores:**

The authors have thoroughly addressed the main concerns from the reviewers. Some reviewers may increase their initial rating to reach the acceptance bar.

---

### Decision · Program_Chairs · 2026-01-26

Accept (Poster)